# Neural Snowflakes: Universal Latent Graph Inference via Trainable Latent Geometries

**Haitz Sáez de Ocáriz Borde**[*]
University of Oxford
Oxford, UK
chri6704@ox.ac.uk

**Anastasis Kratsios**[*]
Department of Mathematics
McMaster University and the Vector Institute
Ontario, Canada
kratsioa@mcmaster.ca

## Abstract

The inductive bias of a graph neural network (GNN) is largely encoded in its specified graph. Latent graph inference relies on latent geometric representations to dynamically rewire or infer a GNN's graph to maximize the GNN's predictive downstream performance, but it lacks solid theoretical foundations in terms of embedding-based representation guarantees. This paper addresses this issue by introducing a trainable deep learning architecture, coined *neural snowflake*, that can adaptively implement fractal-like metrics on $\mathbb{R}^d$. We prove that any given finite weighted graph can be isometrically embedded by a standard MLP encoder, together with the metric implemented by the neural snowflake. Furthermore, when the latent graph can be represented in the feature space of a sufficiently regular kernel, we show that the combined neural snowflake and MLP encoder do not succumb to the curse of dimensionality by using only a low-degree polynomial number of parameters in the number of nodes. This implementation enables a low-dimensional isometric embedding of the latent graph. We conduct synthetic experiments to demonstrate the superior metric learning capabilities of neural snowflakes when compared to more familiar spaces like Euclidean space.

Additionally, we carry out latent graph inference experiments on graph benchmarks. Consistently, the neural snowflake model achieves predictive performance that either matches or surpasses that of the state-of-the-art latent graph inference models. Importantly, this performance improvement is achieved without requiring random search for optimal latent geometry. Instead, the neural snowflake model achieves this enhancement in a differentiable manner.

## 1 Introduction

Geometric deep learning (Bronstein et al., 2017; 2021) is a rapidly developing field that expands the capabilities of deep learning to encompass structured and geometric data, such as graphs, point-clouds, meshes, and manifolds. Graph neural networks (GNNs) derive their knowledge primarily from the specific graph they operate on, but many real-world problems lack an accessible ground truth graph for computation. Latent graph inference aims to address this by dynamically inferring graphs through geometric representations. Existing models lack a strong theoretical foundation and use arbitrary similarity measures for graph inference, lacking principled guidelines. A key challenge is the absence of a differentiable method to deduce geometric similarity for latent graph inference. Recently the concept of neural latent geometry search has been introduced (Sáez de Ocáriz Borde et al., 2023a), which can be formulated as follows: given a search space $\mathfrak{R}$ denoting the set of all possible latent geometries, and the objective function $L_{T,A}(g)$ which evaluates the performance of a given geometry $g$ on a downstream task $T$ for a machine learning model architecture $A$, the objective is to optimize the latent geometry: $\inf_{g \in \mathfrak{R}} L_{T,A}(g)$. In the context of latent graph inference $\mathfrak{R}$ would denote the space of possible geometric similarity measures used to construct the latent graphs. Previous studies have utilized random search to find the optimal geometry in $\mathfrak{R}$ (Kazi et al., 2022; Sáez de Ocáriz Borde et al., 2023c). However, these methods have their limitations as they

---

[*]Equal Contribution.

cannot infer geometry in a differentiable manner, and the representation capabilities of Riemannian manifolds are constrained by certain assumptions inherent in their geometry.

**Contributions.** We introduce a trainable deep learning architecture which we can adaptively implement metrics on $\mathbb{R}^d$ spaces with a fractal-like geometry, called *neural snowflakes*. We prove that together a neural snowflake and a simple MLP encoder are enough to discover any latent graph geometry. In particular, the neural snowflake implements a fractal geometry on $\mathbb{R}^d$ in which any given finite latent weighted graph can be isometrically embedded and the elementary MLP implements that embedding. We show that in cases where the latent weighted graph has favourable geometry, the neural snowflake and MLP encoder break the curse of dimensionality by only requiring a polynomial number of parameters in the graph nodes to implement the isometric embedding. We note the contrast with universal approximation theorems, e.g. Yarotsky (2017); Lu et al. (2021); Shen et al. (2022); Kratsios & Papon (2022), where the number of parameters required to implement a generic approximation depend exponentially on the dimension of the ambient space. Our embedding results exhibit no such exponential dependence on the dimension of the ambient space. We verify our theoretical guarantees experimentally with synthetic metric learning experiments and graph embedding tasks. Additionally we show that the neural snowflake and MLP encoder combination beat or match the state of the art across several latent graph inference benchmarks from the literature. This is achieved by learning the latent geometry in a differentiable manner, utilizing a single model. Thus, the neural snowflake eliminates the need to conduct costly combinatorial searches across numerous combinations of potential embedding spaces.

## 2 BACKGROUND

**Related Work.** In the field of Geometric Deep Learning, most research has relied on human annotators or simple preprocessing algorithms to generate the graph structure used in GNNs. However, even when the correct graph is provided, it may not be optimal for the specific task, and the GNN could benefit from a rewiring process (Topping et al., 2021). Latent graph inference allows models to dynamically learn the intrinsic graph structure of problems where the true graph is unknown (Wang et al., 2019; Kazi et al., 2022). This is particularly relevant in real-world applications where data might only be available in the form of a pointcloud. There are several works in the literature addressing latent graph inference. In particular, we can think of graph rewiring (Arnaiz-Rodríguez et al., 2022; Bi et al., 2022; Guo et al., 2023; Topping et al., 2021) as a subset of latent graph inference in which an input graph is provided to the network, whereas latent graph inference in its most general form allows GNNs to infer a graph starting from only a pointcloud. When the underlying connectivity structure is unknown, traditional architectures like transformers (Vaswani et al., 2017) and attentional multi-agent predictive models (Hoshen, 2017) use a fully-connected graph. This assumption, however, leads to challenges when training with large graphs. Generating sparse graphs can offer computationally tractable solutions (Fetaya et al., 2018) and prevent over-smoothing (Chen et al., 2020a). Various models have been proposed to tackle this problem, starting from Dynamic Graph Convolutional Neural Networks (DGCNNs) (Wang et al., 2019), to approaches that separate graph inference and information diffusion, such as the Differentiable Graph Modules (DGMs) in Cosmo et al. (2020) and Kazi et al. (2022). Recent approaches have focused on generalizing the DGM leveraging product manifolds (Sáez de Ocáriz Borde et al., 2023c;b). Latent graph inference is also referred to as graph structure learning in the literature. A survey of similar methods can be found in Zhu et al. (2021), and some classical methods include LDS-GNN (Franceschi et al., 2019), IDGL (Chen et al., 2020b), and Pro-GNN (Jin et al., 2020). Moreover, recently generalizing latent graph inference to latent topology inference (Battiloro et al., 2023) has also been proposed.

**Graphs.** A weighted graph can be defined as an ordered pair $\mathcal{G} = (V, E, W)$, where $V$ represents a set of nodes (or vertices), $E \subseteq \{\{u, v\} : u, v \in V\}$ forms the collection edges (or links) within the graph, and $W : E \to (0, \infty)$ weights the importance of each edge. An (unweighted) graph $\mathcal{G}$ is a weighted graph for which $W(\{u, v\}) = 1$ for every edge $\{u, v\} \in E$. The neighborhood $\mathcal{N}(v)$ of a node $v \in V$ is the set of nodes sharing an edge with $u$; i.e. $\mathcal{N}(v) \stackrel{\text{def.}}{=} \{u \in V : \{u, v\} \in E\}$.

**Graph Neural Networks.** To compute a message passing *Graph Neural Network* (GNN) layer over a graph $\mathcal{G}$ (excluding edge and graph level features for simplicity), the following equation is typically implemented: $\mathbf{x}_i^{(l+1)} = \varphi\Big(\mathbf{x}_i^{(l)}, \bigoplus_{j \in \mathcal{N}(x_i^{(l)})} \psi(\mathbf{x}_i^{(l)}, \mathbf{x}_j^{(l)})\Big)$. In the given equation, $\psi \in \mathbb{R}^d \times \mathbb{R}^d \to$

$\mathbb{R}^h$ represents a message passing function. The symbol $\bigoplus$ denotes an aggregation function, which must be permutation-invariant, e.g. the sum or max operation. Additionally, $\varphi \in \mathbb{R}^d \times \mathbb{R}^h \to \mathbb{R}^m$ represents a readout function. We note that the update equation is local and relies solely on the neighborhood of the node. Both $\psi$ and $\varphi$ can be Multi-Layer Perceptrons (MLPs). In our manuscript all MLPs will use the $\text{ReLU}(t) \stackrel{\text{def.}}{=} \max\{0, t\}$ activation function, where $t \in \mathbb{R}$. Several special cases have resulted in the development of a wide range of GNN layers: the most well-known being Graph Convolutional Networks (GCNs) (Kipf & Welling, 2017) and Graph Attention Networks (GATs) (Veličković et al., 2018).

**Quasi-Metric Spaces.** While Riemannian manifolds have been employed for formalizing non-Euclidean distances between points, their additional structural properties, such as smoothness and infinitesimal angles, impose substantial limitations, rendering the demonstration of Riemannian manifolds with closed-form distance functions challenging. Quasimetric spaces, isolate the relevant properties of Riemannian distance functions without requiring any of their additional structure for graph embedding. A *quasi-metric space* is a set $X$ with a distance function $d : X \times X \to [0, \infty)$ satisfying for every $x, y, z \in X$: i) $d(x, y) = 0$ if and only if $x = y$, ii) $d(x, y) = d(y, x)$, iii) $d(x, y) \leq C\big(d(x, z) + d(z, y)\big)$, for some constant $C \geq 1$. When $C = 1$, $(X, d)$ is called a metric space, examples include Banach spaces and also, every (geodesic) distance on a Riemannian manifold satisfies (i)-(iii). Conversely, several statistical divergences are weaker structures than quasi-metrics since they fail (ii), and typically fail (iii); see e.g. (Hawkins et al., 2017, Proposition A.2). Property (iii) is called the *C-relaxed triangle inequality* if $C > 1$; otherwise (iii), is called the *triangle inequality*. Quasi-metric spaces typically share many of the familiar properties of metric spaces, such as similar notions of convergence, uniform-continuity of maps between quasimetric spaces, and compactness results for functions between quasimetric spaces such as Arzela-Ascoli theorems (Xia, 2009). The next example of quasimetric spaces are called metric *snowflakes*.

**Example 1** (Xia (2009))**.** *Let $p > 0$ and $(X, d)$ be a metric space. Then, $(X, d^p)$ is a quasimetric space with $C = 2^{p-1}$ if $p > 1$. When $0 < p \leq 1$, then $(X, d^p)$ is a metric space; whence, $C = 1$.*

Snowflakes are a simple tool for constructing new (quasi) metric spaces from old ones with the following properties. Unlike products of Riemannian manifolds, a snowflake's geometry can be completely different than the original untransformed space's geometry. Unlike classical methods for constructing new distances from old ones, e.g. as warped products in differential geometry (Chen, 1999; Alexander & Bishop, 1998), snowflakes admit simple *closed-form* distance functions.

**Proposition 1** (Snowflakes are Metric Spaces - (Weaver, 2018, Proposition 2.50))**.** *Let $f : [0, \infty) \to [0, \infty)$ be a continuous, concave, monotonically increasing function with $f(0) = 0$, and let $(X, d)$ be a metric space; then, $d_f : X \times X \to \mathbb{R}$ is a metric on $X$ $d_f(x, z) \stackrel{\text{def.}}{=} f(d(x, z))$, for any $x, z \in X$.*

## 3 ADAPTIVE GEOMETRIES VIA NEURAL SNOWFLAKES

We overcome one of the main challenges in contemporary Geometric Deep Learning, namely the problem of discovering a latent graph which maximizes the performance of a downstream GNN by searching a catalogue of combinations of products of elementary geometries (Gu et al., 2019); in an attempt to identify which product geometry the latent graph can be best embedded in (Sáez de Ocáriz Borde et al., 2023b;c). The major computational hurdle with these methods is that they pose a *non-differentiable* combinatorial optimization problem with a non-convex objective, making them computationally challenging to scale. Therefore, by designing a class of metrics which are differentiable in their parameters, we can instead discover which geometry best suits a learning task using backpropagation. Core to this is a trainable metric on $\mathbb{R}^d$, defined for any $x, y \in \mathbb{R}^d$ by

$$\|x - y\|_{\sigma_{\alpha,\beta,\gamma,p,C}} \stackrel{\text{def.}}{=} \Big( \underbrace{C_1 \left(1 - e^{-\gamma\|x-y\|}\right)}_{\text{Bounded}} + \underbrace{C_2 \|x - y\|^{\alpha}}_{\text{Fractal}} + \underbrace{C_3 \log(1 + \|x - y\|)^{\beta}}_{\text{Irregular Fractal}} \Big)^{1+|p|} \quad (1)$$

where $0 < \alpha, \beta \leq 1$, $0 \leq p$, $0 \leq C_1, C_2, C_3, \gamma$ not all of which are 0 and $C = (C_i)_{i=1}^3$. The trainable metric in equation 1, coined the *snowflake activation*, is the combination of three components, a *bounded geometry*, a *fractal geometry*, and an *irregular fractal* geometry part; as labeled therein. The first *bounded geometry* can adapt to latent geometries which are bounded akin to spheres, the second *fractal geometry* component can implement any classical snowflake as in Example 1 (where

$0 < p \leq 1$) and the *irregular fractal* adapts to latent geometries much more irregular where the distance between nearby points grows logarithmically at large scales and exponentially at small scales[1]. By Proposition 1, if $p = 0$, the distance in equation 1 is a metric on $\mathbb{R}^d$. For $p > 0$ $(\mathbb{R}^d, \|\cdot\|_{\sigma_{\alpha,\beta,\gamma,p,C}})$ is a quasi-metric space with $2^{p-1}$-relaxed triangle inequality, by Example 1.

### 3.1 NEURAL SNOWFLAKES

We leverage the expressiveness of deep learning, by extending the trainable distance function equation 1 to a deep neural network generating distances on $\mathbb{R}^d$, called the *neural snowflake*.

We begin by rewriting equation 1 as a trainable activation function $\sigma_{a,b} : \mathbb{R} \to [0, \infty)$ which sends on any vector $u \in \mathbb{R}^J$, for $J \in \mathbb{N}_+$, to the $J \times 3$ matrix $\sigma_{a,b}(u)$ whose $j^{th}$ row is

$$\sigma_{a,b}(u)_j \stackrel{\text{def.}}{=} \left(1 - e^{-|u_j|}, |u_j|^a, \log(1 + |u_j|)^b\right). \tag{2}$$

The parameters $0 < a$ and $0 \leq b \leq 1$ are trainable.

We introduce a neural network architecture leveraging the "tensorized" snowflake activation function in equation 2, which can adaptively perturb any metric. To ensure that the neural network model always preserves the metric structure of its input metric, typically the Euclidean metric on $\mathbb{R}^d$, we must constrain the weighs of the hidden layers to ensure that the model satisfies the conditions of Proposition 1. Building on the insights of monotone (Daniels & Velikova, 2010), "input convex" (Amos et al., 2017) neural network architectures, and monotone-value (Weissteiner et al., 2022) neural networks, we simply require that all hidden weights are non-negative and do not all vanish. Lastly, the final layer of our neural snowflake model raises the generated metric to the $(1 + |p|)^{th}$ power as in equation 1. This allows the neural snowflake to leverage the flexibility of quasi-metrics, whenever suitable. They key point here is that by only doing so on the final layer, we can explicitly track the relaxation of the triangle inequality discovered while training. That is, as in Example 1, $C = 2^{p-1}$ if $p > 1$ and $C = 1$ otherwise. Putting it all together, a *neural snowflake* is a map $f : [0, \infty) \to [0, \infty)$, with iterative representation

$$
\begin{aligned}
f(t) &= t_I^{1+|p|} \\
t_i &= B^{(i)} \, \sigma_{a_i,b_i}(A^{(i)} t_{i-1}) \, C^{(i)} \qquad \text{for } i = 1, \ldots, I \\
t_0 &= t
\end{aligned}
\tag{3}
$$

where for $i = 1, \ldots, I$, $A^{(i)}$ is a $\tilde{d}_i \times d_{i-1}$ matrix, $B^{(i)}$ is a $d_i \times \tilde{d}_i$-matrix, and $C^{(i)}$ is a $3 \times 1$ matrix all of which have non-negative weights and at-least one non-zero weight, $p \in \mathbb{R}$; furthermore, for $i = 1, \ldots, I$, $0 < a_i \leq 1$, $0 \leq b_i \leq 1$, $d_1, \ldots, d_I \in \mathbb{N}_+$, and $d_0 = 1 = d_I$. We will always treat the neural snowflake as synonymous with the trainable distance function $\|x - y\|_f \stackrel{\text{def.}}{=} f(\|x - y\|)$, where $x, y \in \mathbb{R}^d$ for some contextually fixed $d$ and $f$ is as in equation 3.

## 4 INFERABILITY GUARANTEES

This section contains the theoretical guarantees underpinning the neural snowflake graph inference model. We first show that it is universal, in the sense of graph representation learning, which we formalize. We then derive a series of qualitative guarantees showing that the neural snowflake graph inference model requires very few parameters to infer any latent weighted graph. In particular, neural snowflakes require a computationally feasible number of parameters to be guaranteed to work.

### 4.1 UNIVERSAL GRAPH EMBEDDING

Many graph inference pipelines depend on preserving geometry representations or encodings within latent geometries when inferring the existence of an edge between any two points (nodes) in a point cloud. Therefore, the effectiveness of any algorithm in this family of encoders hinges on its capacity to accurately or approximately represent the geometry of the latent graph. In this work, we demonstrate that the neural snowflake can infer any latent graph in this way. Thus, we formalize what

---

[1]Note that $\log(1 + \|x - y\|) \approx 1 - e^{-\|x-y\|}$ when $0 \approx \|x - y\|$.

it means for a *graph inference model* to be able to represent any latent (weighted) graph structure in $\mathbb{R}^D$ based on a class of geometries $\mathfrak{R}$. For any $D \in \mathbb{N}_+$, we call a pair $(\mathfrak{E}, \mathfrak{R})$ a *graph inference model*, on $\mathbb{R}^D$, if $\mathfrak{R}$ is a family of quasi-metric spaces, and $\mathfrak{E}$ is a family of maps with domain $\mathbb{R}^D$ and codomain in some member $(\mathcal{R}, d_\mathcal{R})$ of $\mathfrak{R}$. Whenever $\mathbb{R}^D$ is clear from the context, we do not explicitly mention it.

### 4.1.1 UNIVERSAL RIEMANNIAN REPRESENTATION IS IMPOSSIBLE

Our primary qualitative guarantee asserts the universality of the graph inference model $(\mathfrak{E}, \mathfrak{R})$, where $\mathfrak{E}$ represents the set of MLPs into $\mathbb{R}^d$ with ReLU activation functions, and $\mathfrak{R}$ comprises all $(\mathbb{R}^d, | \cdot |_f)$, where $f$ is a neural snowflake; for integers $d \in \mathbb{N}_+$. We now formalized what is meant by a *universal* graph embedding model.

**Definition 1** (Universal Graph Embedding). *A graph inference model $(\mathfrak{E}, \mathfrak{R})$ is universal if: for every non-empty finite subset $V \subseteq \mathbb{R}^D$ and every connected weighted graph $\mathcal{G} = (V, E, W)$ there is a (quasi-metric) representation space $(\mathcal{R}, d_\mathcal{R}) \in \mathfrak{R}$ and an encoder $\mathcal{E} : \mathbb{R}^D \to \mathcal{R}$ in $\mathfrak{E}$ satisfying*

$$d_\mathcal{G}(u, v) = d_\mathcal{R}(\mathcal{E}(u), \mathcal{E}(v)) \qquad \forall\, u, v \in V.$$

Our interest in universal graph inference models lies in their ability to infer graph edges. This is done by first learning an embedding $\mathcal{E} \in \mathfrak{E}$ into some representation space $(\mathcal{R}, d_\mathcal{R}) \in \mathfrak{R}$ and subsequently sampling edges based on nearest neighbors within the aforementioned embedding.

One technical point worth noting is that, when forming sets of nearest neighbors, ties between equidistant points are broken arbitrarily. This is accomplished by indexing (possibly randomly) the graph's vertices and selecting the first few nearest points based on the ordering of that index, similar to the approach in Fakcharoenphol et al. (2004).

The formalization of this reconstruction procedure, in Theorem 1, uses the following notation. For every positive integer $N$, we denote the first $N$ positive integers by $[N] \overset{\text{def.}}{=} \{1, \dots, N\}$. For every quasi-metric (representation) space $(\mathcal{R}, d_\mathcal{R})$ each point $x \in \mathcal{R}$, and each radius $r \geq 0$ the closed unit ball about $x$ of radius $r$ is $\bar{B}_\mathcal{R}(x, r) \overset{\text{def.}}{=} \{u \in \mathcal{R} : d_\mathcal{R}(x, y) \leq r\}$.

**Theorem 1** (Generic Graph Reconstruction via Universal Graph Inference Models). *Fix $D \in \mathbb{N}_+$ and a latent graph inference model $(\mathfrak{E}, \mathfrak{R})$ on $\mathbb{R}^D$. For every non-empty finite subset $V \subseteq \mathbb{R}^D$, every graph $G = (V, E)$, and each index $V = \{v_i\}_{i=1}^N$ there exists: a quasi-metric (representation) space $(\mathcal{R}, d_\mathcal{R}) \in \mathfrak{R}$ and an encoder $\mathcal{E} : \mathbb{R}^D \to \mathcal{R}$ in $\mathfrak{E}$ such that: for each $i \in [N]$ there is a (number of nearest neighbours) $k_i \in [N]$ satisfying*

$$\{u_i, u_j\} \in E \Leftrightarrow j \leq i^\star \text{ and } d_\mathcal{R}\big(\mathcal{E}(u_i), \mathcal{E}(u_j)\big) \leq r(k_i)$$

*where $r(k_i) \overset{\text{def.}}{=} \inf \big\{r \geq 0 : \#\{v \in V : d_\mathcal{R}\big(\mathcal{E}(u_i), \mathcal{E}(v)\big) \leq r\} \geq k_i\big\}$ and where $i^\star \overset{\text{def.}}{=} \{j \in [N] : \#(\bar{B}_\mathcal{R}(u_i, r(k_i)) \cap \{u_s\}_{s=1}^j) \leq k_i\}$.*

Theorem 1 shows that if a latent graph inference model is universal, then it can be used to reconstruct the edge set of any latent graph structure by first embedding the vertices/point-cloud into a latent representation space and then joining nearest neighbours. The next natural question is: *"How does one build a universal latent graph inference model which is differentiable?"*

It is known that the family of Euclidean spaces $\mathfrak{R} = \{(\mathbb{R}^d, \| \cdot \|)\}_{d \in \mathbb{N}_+}$ are not flexible enough to isometrically accommodate all weighted graphs; even if $\mathfrak{E}$ is the family of *all functions* from any $\mathbb{R}^D$ into any Euclidean space $(\mathbb{R}^d, \| \cdot \|)$. This is because, some weighted graphs do not admit isometric embeddings into any Euclidean space (Bourgain et al., 1986; Linial et al., 1995; Matoušek, 1997). Even infinite-dimensions need not be enough, since for every $n \in \mathbb{N}_+$, there is an $n$-point weighted graph which cannot be embedded in the Hilbert space $\ell^2$ with distortion less than $\Omega(\log(n)/\log(\log(n)))$ (Bourgain, 1985, Proposition 2). In particular, it cannot be isometrically embedded therein. For example, any $l$-leaf tree embeds in $d$-dimensional Euclidean space with distortion at-least $\Omega(l^{1/d})$. In contrast, any finite tree can embed with arbitrary low-distortion into the hyperbolic plane (Kratsios et al., 2023b), which is a particular two-dimensional non-flat Riemannian geometry. Several authors (Sarkar, 2012) have shown that cycle graphs can be embedded isometrically in spheres of appropriate dimension and radius (Schoenberg, 1942), or in the product

of spheres (Guella et al., 2016), but they cannot be embedded isometrically in Euclidean space (Enflo, 1976). These observations motivate geodesic deep learners (Liu et al., 2019; Chamberlain et al., 2017; Chami et al., 2019; Sáez de Ocáriz Borde et al., 2023c;b) and network scientists (Verbeek & Suri, 2014) to use families of Riemannian representation spaces $\mathfrak{R}$ in which it is hoped that general graphs can faithfully be embedded, facilitating embedding-based latent graph inference. Unfortunately, 5 nodes and 5 edges are enough to construct a graph which cannot be isometrically embedded into *any* non-pathological Riemannian manifold.

**Proposition 2** (Riemannian Representation Spaces are Too Rigid to be Universal). *For any $D \in \mathbb{N}_+$ and any 5-point subset $V$ of $\mathbb{R}^D$, there exists a set of edges $E$ on $V$ such that:*

    *(i) the graph $\mathcal{G} \stackrel{\text{def.}}{=} (V, E)$ is connected*

    *(ii) for every complete[2] and connected smooth Riemannian manifold $(\mathcal{R}, g)$ there does not exist an isometric embedding $\varphi : (V, d_\mathcal{G}) \to (\mathcal{R}, d_\mathcal{R})$*

*where $d_\mathcal{G}$ and $d_\mathcal{R}$ respectively denote the shortest path (geodesic) distances on $\mathcal{G}$ and on $(\mathcal{R}, g)$.*

In other words, Proposition 2 shows that if $\mathfrak{R}$ is any set of non-pathological Riemannian manifolds and $\mathfrak{E}$ any set of functions from $\mathbb{R}^D$ into any Riemannian manifold $(\mathcal{R}, g)$ in $\mathfrak{R}$ the graph inference model $(\mathfrak{E}, \mathfrak{R})$ is not universal. Furthermore, the graph "breaking its universality" is nothing obscure but a simple 5 node graph. Note that, no edge weights (not equal to 1) are needed in Proposition 2.

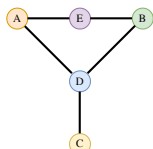

*Figure 1:* **Explanation of Proposition 2**: The Graph of Proposition 2 cannot be isometrically embedded into any complete and connected (smooth) Riemannian manifold. Briefly, the issue is that any isometric embedding into such a Riemannian manifold must exhibit a pair of geodesics one of which travels from the embeddings of node $C$ to node $A$, while passing through the embedding of node $D$; and likewise, the other of which travels from the embedding of node $C$ to node $B$ and again passes through the embedding of node $D$. However, this would violate the local uniqueness of geodesics in such a Riemannian manifold, around the embedding of node $D$ (implied by the Picard-Lindelöf theorem for ODEs); thus no such embedding can exist.

Proposition 2 improves on (Kratsios et al., 2023a, Propositions 13 and 15) since the latter only show that no compact connected Riemannian manifolds (e.g. products of spheres) and no connected Riemannian manifold with bounded non-positive sectional curvatures (e.g. products of hyperbolic spaces) can accommodate certain sequences of expander graphs (see (Kratsios et al., 2023a, Remark 14)). However, those results do not rule out more complicated Riemannian representation spaces; e.g. the products of spheres, hyperbolic, and Euclidean spaces recently explored by Gu et al. (2019); Tabaghi et al. (2021); Di Giovanni et al. (2022); Sáez de Ocáriz Borde et al. (2023c).

### 4.1.2 UNIVERSAL REPRESENTATION IS POSSIBLE WITH NEURAL SNOWFLAKE

Juxtaposed against Proposition 2, our first main result shows that together, neural snowflake and MLPs, are universal graph embedding models.

**Theorem 2** (Universal Graph Embedding). *Let $D \in \mathbb{N}_+$ and $\mathcal{G}$ be a weighted graph with $V \subseteq \mathbb{R}^D$ with $I \in \mathbb{N}_+$ vertices. There exists an embedding dimension $d \in \mathbb{N}_+$, an MLP $\mathcal{E} : \mathbb{R}^D \to \mathbb{R}^{\bar{d}}$ with* ReLU *activation function, and a neural snowflake $f$ such that*

$$d_\mathcal{G}(u, v) = \|\mathcal{E}(u) - \mathcal{E}(v)\|_f,$$

*for each $u, v \in V$. The $(\mathbb{R}^d, \|\cdot\|_f)$ supports a $2^{O(\log(1 + \frac{1}{I-1})^{-1})}$-relaxed triangle inequality. If $\mathcal{G}$ is a tree then $(\mathbb{R}^d, \|\cdot\|_f)$ instead supports an 8-triangle inequality.*

**Comparison: State-of-the-Art Deep Embedding Guarantees.** Recently, Kratsios et al. (2023a) built on Andoni et al. (2018) and proposed a universal graph embedding model which uses the

---

[2]Here, complete is meant in the sense of metric spaces; i.e. all Cauchy sequences in a complete metric space have a limit therein.

transformer architecture of Kratsios (2021) to represent graphs in the order 2-Wasserstein space on $\mathbb{R}^3$. The drawbacks of this approach are that the metric is not available in closed-form, it is computationally infeasible to evaluate exactly for large graph embeddings, and it is still challenging to evaluate approximately (Cuturi, 2013). In contrast, Theorem 2 guarantees that a simple MLP can isometrically embedding any weighted graph into a finite-dimensional representation space with closed-form distance function explicitly implemented by a neural snowflake.

**Comparison: MLP without Neural Snowflake.** We examine the necessity of the neural snowflake in Theorem 2, by showing that the MLP alone cannot isometrically represent any weighted graph into its natural output space; namely some Euclidean space.

**Theorem 3** (Neural Snowflakes & MLPs Are More Powerful For Representation Learning Than MLPs). *Let $d, D \in \mathbb{N}_+$. The following hold:*

  (i) **No Less Expressive Than MLP:** *For any weighted graph $\mathcal{G} = (V, E, W)$ with $V \subseteq \mathbb{R}^D$, if there is an MLP $\mathcal{E} : \mathbb{R}^D \to \mathbb{R}^d$ which isometrically embeds $\mathcal{G}$ then there is a neural snowflake $f$ and an MLP $\mathcal{E} : \mathbb{R}^D \to \mathbb{R}^d$ which isometrically embeds $\mathcal{G}$ into $(\mathbb{R}^d, \|\cdot\|_f)$.*

  (ii) **Strictly More Expressive Than MLP:** *There exists a complete weighted graph $G = (V, E, W)$ with $V \subset \mathbb{R}^D$ by any MLP $\tilde{\mathcal{E}} : \mathbb{R}^D \to \mathbb{R}^d$ but for which there exists a neural snowflake $f$ and an MLP $\mathcal{E} : \mathbb{R}^D \to \mathbb{R}^d$ that isometrically embeds $\mathcal{G}$ into $(\mathbb{R}^D, \|\cdot\|_f)$.*

## 4.2 ISOMETRIC REPRESENTATION GUARANTEES - BY SMALL NEURAL SNOWFLAKES

Theorem 2 offers a qualitative assurance that the neural snowflake can represent any finite weighted graph. We now show that any weighted graph which can be isometrically represented in the latent geometry induced by kernel regressors, can be implemented by a neural snowflake. We assume that the latent geometry of the weighted graph is encoded in a low-dimensional space and the distances in that low-dimensional space are given by a *radially symmetric* and positive-definite kernel.

**Assumption 1** (Latent Radially-Symmetric Kernel). *There are $d, D \in \mathbb{N}_+$, a feature map $\Phi : \mathbb{R}^D \to \mathbb{R}^d$, and a non-constant positive-definite function[3] $f : [0, \infty) \to [0, \infty)$ satisfying*

$$d_{\mathcal{G}}(x, y) = \bar{f}\big(\|\Phi(x) - \Phi(y)\|\big),$$

*for every $x, y \in \mathbb{R}^d$; where $\bar{f}(t) \overset{\text{def.}}{=} f(0) - f(t)$.*

Several satisfy Assumption 1, as we emphasize using two exotic examples (Appendix D).

**Example 2.** *The map $f(t) \overset{\text{def.}}{=} (1 + \sum_{k=1}^K |t|^{r_k})^{-\beta}$ satisfies Assumption 1 for any $K \in \mathbb{N}_+$ and $0 \leq r_1, \ldots, r_K, \beta \leq 1$.*

**Example 3.** *The map $f(t) \overset{\text{def.}}{=} \exp\big(\frac{-a(t-1)}{\log(t)}\big)$ satisfies Assumption 1, for all $a > 0$.*

We find that the neural snowflake can implement an isometric representation of the latent geometry, as in Theorem 2, using a small number of parameters comparable to Theorem 5.

**Theorem 4** (Quantitative Embedding Guarantees for Bounded Metric Geometries). *Let $D, d \in \mathbb{N}_+$, $G$ be a finite weighted graph with $V \subset \mathbb{R}^D$ and suppose that Assumption 1 (or equation 2) holds. Then, there is a neural snowflake $(\mathcal{E}, f)$ satisfying*

$$\|\Phi(v) - \Phi(u)\|_f = \|\mathcal{E}(v) - \mathcal{E}(u)\|_{\bar{f}}$$

*for every $u, v \in V$. Furthermore, the depth and width of $\mathcal{E}$ and $f$ are recorded in Table 1.*

## 4.3 REPRESENTATION GUARANTEES LEVERAGING DISTORTION

Theorem 2 shows the neural snowflake and an simple MLP are *universal graph embedding* models. By allowing the neural snowflake and MLP some possible slack to distort the latent graph's geometry, either by stretching or contracting pairwise distances ever so slightly, we are able to derive explicit bounds on the embedding dimension $d$ and on the complexity of the neural snowflake and MLP.

---

[3] A positive-definite function, is map $f : \mathbb{R} \to \mathbb{C}$ for which each $\big(f(x_i - x_j)\big)_{i,j=1}^N$ is a positive-definite matrix, for every $N \in \mathbb{N}_+$ and each $x_1, \ldots, x_N \in \mathbb{R}$.

**Theorem 5** (Quantitative Approximately Isometric Embeddings for Weighted Graphs). *Let $G = (V, E, W)$ be a weighted graph $N \in \mathbb{N}_+$ nodes, a non-negative weighting function $W$, and $V \subset \mathbb{R}^d$. For every $1 < p < 2$, there exists an MLP with ReLU activation function $\mathcal{E} : \mathbb{R}^d \to \mathbb{R}^{O(\log(\dim(G)))}$ and a snowflaking network $f : \mathbb{R} \to \mathbb{R}$ such that: for every $x, u \in V$*

$$d_G(x, u) \le \|\mathcal{E}(x) - \mathcal{E}(u)\|_f \le D^p d_G(x, u),$$

*where $D \in \mathcal{O}\left(\frac{\log(\dim(G))^2}{p^2}\right)$ and $(\mathbb{R}^n, \|\cdot\|_f)$ is a quasi-metric space satisfying the $2^{p-1}$-triangle inequality. The depth and width of $\mathcal{E}$ and $f$ are recorded in Table 1.*

Table 1: Complexity of the Neural Snowflake and MLPs. Here, $\mathcal{O}$ suppresses a constant depending only on $D$ and $C_{\mathcal{G}} > 0$ depends only on $\mathcal{G}$ (explicit constants are given in Appendix B).

| Geometry | Net. | Hidden Layers (Depth $-1$) | Width | Theorem |
|---|---|---|---|---|
| General | $f$ | $\Theta(1)$ | $\Theta(1)$ | 5 |
| Bounded | $f$ | $\Theta(1)$ | $\mathcal{O}(I^2)$ | 4 |
| All | $\mathcal{E}$ | $\mathcal{O}(I\sqrt{I \log(I)} \log(I^2 C_{\mathcal{G}}))$ | $\mathcal{O}(DI + d)$ | 4 and 5 |

**Discussion - Comparison with the Best Proven Deep Representation.** The complexity of the neural snowflake and MLP are reported in Table 1. Theorem 5 shows that the neural snowflake can match the best known embedding guarantees by a deep learning model with values in a curved infinite-dimensional space (Kratsios et al., 2023a, Theorem 4), both in terms of distortion and the number of parameters. The neural snowflake's adaptive geometry allows for the representations to be implemented in $\mathcal{O}(\log(\dim(G))$ dimensions and the implemented representation space $(\mathbb{R}^d, \|\cdot\|_f)$ has an explicit closed-form distance function making it trivial to evaluate; unlike the distance function in the representation space of Kratsios et al. (2023a). Our guarantees show that neural snowflakes can theoretically represent any weighted graph; either isometrically or nearly isometrically with provable few parameters.

## 5 EXPERIMENTAL RESULTS

Next, we validate our embedding results through synthetic graph experiments, and we demonstrate how the combined representation capabilities of neural snowflakes and MLPs can enhance existing state-of-the-art latent graph inference pipelines.

**Synthetic Embedding Experiments: Neural Snowflakes vs Euclidean Graph Embedding Spaces.** To assess the effectiveness of neural snowflakes as well as to compare their performance with that of MLPs in approximating metrics using Euclidean space, we conduct synthetic graph embedding experiments. Specifically, we focus on fully connected graphs, where the node coordinates are randomly sampled from a multivariate Gaussian distribution within a 100-dimensional hypercube in Euclidean space, denoted as $\mathbb{R}^{100}$. The weights of the graphs are computed according to the metrics in Table 2. In the leftmost column, the MLP model projects the node features in $\mathbb{R}^{100}$ to $\mathbb{R}^2$ and aims at approximating the edge weights of the graph using the euclidean distance $\|\mathrm{MLP}(\mathbf{x}) - \mathrm{MLP}(\mathbf{y})\|$ between the projected features. In the central column, the neural snowflake $f$ learns a quasi-metric based on the input Euclidean metric previously mentioned in $\mathbb{R}^2$, $f(\|\mathrm{MLP}(\mathbf{x}) - \mathrm{MLP}(\mathbf{y})\|)$. Finally, in the rightmost column the neural snowflake learns based on the Euclidean distance between features in $\mathbb{R}^{100}$: $f(\|\mathbf{x} - \mathbf{y}\|)$. In order to demonstrate the superior embedding capabilities of neural snowflakes compared to Euclidean spaces, we intentionally equip the MLP in the first column with a significantly larger number of model parameters than the other models. This is done to highlight the fact that, despite having fewer learnable parameters, neural snowflakes outperform Euclidean spaces by orders of magnitude. Furthermore, the results of our experiments reveal that neural snowflakes exhibit remarkable flexibility in learning metrics even in lower-dimensional spaces such as $\mathbb{R}^2$. This is evident from the similarity of results obtained in the rightmost column (representing $\mathbb{R}^{100}$) compared to those achieved in the case of learning the metric in $\mathbb{R}^2$. Additional information regarding these embedding experiments can be found in Appendix I.

**Graph Benchmarks.** In this section, we present our findings on using Neural Snowflakes compared to other latent graph inference models. Our objective is to evaluate the representation power offered

*Table 2:* Results for synthetic graph embedding experiments for the test set. The Neural Snowflake models are able to learn the metric better with substantially lesser number of model parameters.

| | MLP | Neural Snowflake (+ MLP) | Neural Snowflake |
|---|---|---|---|
| No. Parameters | 5422 | 4169 | 847 |
| Embedding space, $\mathbb{R}^n$ | 2 | 2 | 100 |
| Metric | Mean Square Embedding Error | | |
| $\|\mathbf{x} - \mathbf{y}\|^{0.5} \log(1 + \|\mathbf{x} - \mathbf{y}\|)^{0.5}$ | 1.3196 | 0.0034 | **0.0029** |
| $\|\mathbf{x} - \mathbf{y}\|^{0.1} \log(1 + \|\mathbf{x} - \mathbf{y}\|)^{0.9}$ | 1.2555 | 0.0032 | **0.0032** |
| $1 - \frac{1}{(1+\|\mathbf{x}-\mathbf{y}\|^{0.5})}$ | 0.1738 | **0.00004** | **0.00004** |
| $1 - \exp\frac{-(\|\mathbf{x}-\mathbf{y}\|-1)}{\log(\|\mathbf{x}-\mathbf{y}\|)}$ | 0.3249 | 0.00008 | **0.00008** |
| $1 - \frac{1}{(1+\|\mathbf{x}-\mathbf{y}\|)^{0.2}}$ | 0.0315 | **0.00005** | 0.00005 |
| $1 - \frac{1}{1+\|\mathbf{x}-\mathbf{y}\|^{0.2}+\|\mathbf{x}-\mathbf{y}\|^{0.5}}$ | 0.2484 | **0.00002** | **0.00002** |

by various latent space metric spaces, and thus, we contrast our results with the original DGM (Kazi et al., 2022) and its non-Euclidean variants (Sáez de Ocáriz Borde et al., 2023b;c). We also introduce a new variant of the DGM which uses a snowflake activation on top of the classical DGM, to equip the module with a snowflake quasi-metric space. We ensure a fair evaluation, by conducting all experiments using the same GCN model and only modify the latent geometries used for latent graph inference. We take care to maintain consistency in the number of model parameters, as well as other training specifications such as learning rates, training and testing splits, number of GNN layers, and so on. This approach guarantees that all comparisons are solely based on the metric (or quasi-metric) space utilized for representations. By eliminating the influence of other factors, we can obtain reliable and trustworthy experimental results. A detailed and systematic analysis of the results is provided in Appendix I.

We first present results from latent graph inference on the well-known Cora and CiteSeer homophilic graph benchmarks. We use a consistent latent space dimensionality of 8 and perform the Gumbel top-k trick for edge sampling with a k value of 7. The models all share the same latent space dimensionality, differing solely in their geometric characteristics. In scenarios where a product manifold is used, the overall manifold is constructed by amalgamating two 4-dimensional manifolds through a Cartesian product. This yields a total latent space dimensionality of 8. This methodology ensures an equitable comparison based exclusively on geometric factors. All other parameters, comprising network settings and training hyperparameters, remain unaltered. For all DGM modules, GCNs are employed as the underlying GNN diffusion layers. Table 3 displays the results for Cora and CiteSeer, leveraging the original dataset graphs as inductive biases.

*Table 3:* Results for Cora and CiteSeer leveraging the original input graph as an inductive bias.

| | | Cora | CiteSeer |
|---|---|---|---|
| Model | Metric Space | Accuracy (%) $\pm$ Standard Deviation | |
| Neural Snowflake | Snowflake | **87.07**$_{\pm 3.45}$ | **74.76**$_{\pm 1.74}$ |
| DGM | Snowflake | 85.41$_{\pm 3.70}$ | **74.19**$_{\pm 2.08}$ |
| DGM | Euclidean | **85.77**$_{\pm 3.64}$ | 73.67$_{\pm 2.30}$ |
| DGM | Hyperboloid | 85.25$_{\pm 3.34}$ | 73.46$_{\pm 1.79}$ |
| DGM | Poincare | **86.07**$_{\pm 3.53}$ | 71.23$_{\pm 5.53}$ |
| DGM | Spherical | 76.14$_{\pm 2.84}$ | 73.13$_{\pm 2.93}$ |
| DGM | Euclidean $\times$ Hyperboloid | 84.33$_{\pm 2.56}$ | 73.29$_{\pm 2.18}$ |
| DGM | Hyperboloid $\times$ Hyperboloid | 84.59$_{\pm 5.40}$ | **74.42**$_{\pm 1.83}$ |
| GCN | Euclidean | 83.50$_{\pm 2.00}$ | 70.03$_{\pm 2.04}$ |

Next we perform experiments on Cora and CiteSeer without considering their respective graphs, that is, the latent graph inference models only take pointclouds as inputs in this case. We also include results for the Tadpole and Aerothermodynamics datasets used in Sáez de Ocáriz Borde et al. (2023c). Note that in these experiments all models used GCNs and a fixed latent space dimensionality of 8, unlike in the original paper which uses a larger latent spaces and GAT layers. The effect of changing the latent space dimensionality is further explored in Appendix I. For both Tadpole and Aerothemodynamics, the Gumbel Top-k algorithm samples 3 edges per node. See Table 4.

*Table 4:* Results for Cora and CiteSeer, and the real-world Tadpole and Aerothermodynamics datasets, without leveraging the original input graph as an inductive bias.

| | | Cora | CiteSeer | Tadpole | Aerothermodynamics |
|---|---|---|---|---|---|
| Model | Metric Space | Accuracy (%) $\pm$ Standard Deviation | | | |
| Neural Snowflake | Snowflake | $71.22_{\pm4.27}$ | $67.80_{\pm2.44}$ | $90.02_{\pm3.51}$ | $88.65_{\pm3.10}$ |
| DGM | Snowflake | $69.51_{\pm4.42}$ | $66.86_{\pm2.82}$ | $91.61_{\pm4.38}$ | $88.55_{\pm2.35}$ |
| DGM | Euclidean | $68.37_{\pm5.39}$ | $68.10_{\pm2.80}$ | $89.29_{\pm4.66}$ | $88.28_{\pm2.61}$ |
| DGM | Hyperboloid | $70.00_{\pm4.08}$ | $68.34_{\pm1.59}$ | $88.75_{\pm5.11}$ | $88.29_{\pm2.85}$ |
| DGM | Poincare | $65.74_{\pm4.02}$ | $64.63_{\pm2.98}$ | $86.96_{\pm5.30}$ | $89.00_{\pm2.34}$ |
| DGM | Spherical | $37.03_{\pm14.33}$ | $20.00_{\pm4.13}$ | $82.32_{\pm11.74}$ | $88.37_{\pm3.00}$ |
| DGM | Euclidean $\times$ Hyperboloid | $62.18_{\pm6.61}$ | $65.72_{\pm2.48}$ | $90.71_{\pm3.17}$ | $88.20_{\pm3.23}$ |
| DGM | Hyperboloid $\times$ Hyperboloid | $67.14_{\pm4.19}$ | $64.33_{\pm9.44}$ | $89.64_{\pm5.57}$ | $89.55_{\pm2.52}$ |
| MLP | Euclidean | $58.92_{\pm3.28}$ | $59.48_{\pm2.14}$ | $87.70_{\pm3.46}$ | $80.99_{\pm8.34}$ |

From the results, we can see that models using snowflake metric spaces are consistently amongst the top performers for both Cora and CiteSeer. On the other hand, employing non-learnable metric spaces necessitates conducting an exploration of various latent space geometries to achieve optimal outcomes, since there is no metric space that consistently outperforms the rest regardless of the dataset. In our synthetic experiments, we have clearly demonstrated the remarkable advantage of neural snowflakes in learning metric spaces with enhanced flexibility, when compared to Euclidean space. In the context of latent graph inference, this distinction is not as pronounced as observed in the synthetic experiments. This can be attributed to the suboptimal nature of the Gumbel Top-k edge sampling algorithm (Appendix E), a topic discussed in other research works (Kazi et al., 2022; Battiloro et al., 2023), which essentially introduces a form of "distortion" to the learned metric. Yet, it is worth noting that the development of improved edge sampling algorithms to foster better synergy between metric space learning and graph construction is not the primary focus of this paper. Instead, our emphasis is on introducing a more comprehensive and trainable metric space and integrating it with existing edge sampling techniques.

## 6 CONCLUSION

Our theoretical analysis showed that a small neural snowflake, denoted as $f$, can adaptively implement fractal-like geometries $|\cdot|_f$ on $\mathbb{R}^d$, which are flexible enough to grant a small MLP the capacity to isometrically embed any finite graph. We showed that the non-smooth geometry implemented by the neural snowflake is key by showing that there are simple graphs that cannot be isometrically embedded into any smooth Riemannian representation space. We then explored several cases in which the combination of neural snowflakes and MLPs requires a small number of total parameters, independent of ambient dimensions, to represent certain classes of regular weighted graphs. We complemented our theoretical analysis by extensively exploring the best approaches to implement neural snowflakes, ensuring stability during training, in both synthetic and graph benchmark experiments. We also introduced a snowflake activation that can easily be integrated into the DGM module using a differentiable distance function, enabling the DGM to leverage a snowflake quasi-metric. We conducted tests on various graph benchmarks, systematically comparing the effectiveness of snowflake quasi-metric spaces for latent graph inference with other Riemannian metrics such as Euclidean, hyperbolic variants, spherical spaces, and product manifolds of model spaces. Our proposed model is consistently able to match or outperform the baselines while learning the latent geometry in a differentiable manner and without having to perform random search to find the optimal embedding space.

Note that our experiments were conducted in accordance with the differentiable graph module framework for discrete edge sampling as proposed by Kazi et al. (2022). Our primary objective was to compare the representation capabilities of various (quasi-)metric spaces, while keeping all other model architecture choices constant. Recently, the NodeFormer (Wu et al., 2023) architecture was introduced, enabling the scalability of latent graph inference to large graphs. However, this development is slightly tangential to the research presented in this work, which focuses on analyzing the geometric characteristics of different embedding spaces. We propose considering the incorporation of the geometric notions discussed in this work into new scalable architectures, such as the NodeFormer, as part of future research.

# 7 ACKNOWLEDGMENT AND FUNDING

AK acknowledges financial support from the NSERC Discovery Grant No. RGPIN-2023-04482 and their McMaster Startup Funds. The authors would also like to thank Giulia Livieri for her helpful feedback.

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
