\textbf{4.27}}$ | **67.80**$_{\pm\textbf{2.44}}$ | **90.02**$_{\pm\textbf{3.51}}$ | **88.65**$_{\pm\textbf{3.10}}$ |
| DGM | Snowflake | **69.51**$_{\pm\textbf{4.42}}$ | 66.86$_{\pm2.82}$ | **91.61**$_{\pm\textbf{4.38}}$ | 88.55$_{\pm2.35}$ |
| DGM | Euclidean | 68.37$_{\pm5.39}$ | **68.10**$_{\pm\textbf{2.80}}$ | 89.29$_{\pm4.66}$ | 88.28$_{\pm2.61}$ |
| DGM | Hyperboloid | **70.00**$_{\pm\textbf{4.08}}$ | **68.34**$_{\pm\textbf{1.59}}$ | 88.75$_{\pm5.11}$ | 88.29$_{\pm2.85}$ |
| DGM | Poincare | 65.74$_{\pm4.02}$ | 64.63$_{\pm2.98}$ | 86.96$_{\pm5.30}$ | **89.00**$_{\pm\textbf{2.34}}$ |
| DGM | Spherical | 37.03$_{\pm14.33}$ | 20.00$_{\pm4.13}$ | 82.32$_{\pm11.74}$ | 88.37$_{\pm3.00}$ |
| DGM | Euclidean $\times$ Hyperboloid | 62.18$_{\pm6.61}$ | 65.72$_{\pm2.48}$ | **90.71**$_{\pm\textbf{3.17}}$ | 88.20$_{\pm3.23}$ |
| DGM | Hyperboloid $\times$ Hyperboloid | 67.14$_{\pm4.19}$ | 64.33$_{\pm9.44}$ | 89.64$_{\pm5.57}$ | **89.55**$_{\pm\textbf{2.52}}$ |
| MLP | Euclidean | 58.92$_{\pm3.28}$ | 59.48$_{\pm2.14}$ | 87.70$_{\pm3.46}$ | 80.99$_{\pm8.34}$ |

From the results, we can see that models using snowflake metric spaces are consistently amongst the top performers for both Cora and CiteSeer. On the other hand, employing non-learnable metric spaces necessitates conducting an exploration of various latent space geometries to achieve optimal outcomes, since there is no metric space that consistently outperforms the rest regardless of the dataset. In our synthetic experiments, we have clearly demonstrated the remarkable advantage of neural snowflakes in learning metric spaces with enhanced flexibility, when compared to Euclidean space. In the context of latent graph inference, this distinction is not as pronounced as observed in the synthetic experiments. This can be attributed to the suboptimal nature of the Gumbel Top-k edge sampling algorithm (Appendix E), a topic discussed in other research works (Kazi et al., 2022; Battiloro et al., 2023), which essentially introduces a form of "distortion" to the learned metric. Yet, it is worth noting that the development of improved edge sampling algorithms to foster better synergy between metric space learning and graph construction is not the primary focus of this paper. Instead, our emphasis is on introducing a more comprehensive and trainable metric space and integrating it with existing edge sampling techniques.

## 6 CONCLUSION

Our theoretical analysis showed that a small neural snowflake, denoted as $f$, can adaptively implement fractal-like geometries $|\cdot|_f$ on $\mathbb{R}^d$, which are flexible enough to grant a small MLP the capacity to isometrically embed any finite graph. We showed that the non-smooth geometry implemented by the neural snowflake is key by showing that there are simple graphs that cannot be isometrically embedded into any smooth Riemannian representation space. We then explored several cases in which the combination of neural snowflakes and MLPs requires a small number of total parameters, independent of ambient dimensions, to represent certain classes of regular weighted graphs. We complemented our theoretical analysis by extensively exploring the best approaches to implement neural snowflakes, ensuring stability during training, in both synthetic and graph benchmark experiments. We also introduced a snowflake activation that can easily be integrated into the DGM module using a differentiable distance function, enabling the DGM to leverage a snowflake quasi-metric. We conducted tests on various graph benchmarks, systematically comparing the effectiveness of snowflake quasi-metric spaces for latent graph inference with other Riemannian metrics such as Euclidean, hyperbolic variants, spherical spaces, and product manifolds of model spaces. Our proposed model is consistently able to match or outperform the baselines while learning the latent geometry in a differentiable manner and without having to perform random search to find the optimal embedding space.

Note that our experiments were conducted in accordance with the differentiable graph module framework for discrete edge sampling as proposed by Kazi et al. (2022). Our primary objective was to compare the representation capabilities of various (quasi-)metric spaces, while keeping all other model architecture choices constant. Recently, the NodeFormer (Wu et al., 2023) architecture was introduced, enabling the scalability of latent graph inference to large graphs. However, this development is slightly tangential to the research presented in this work, which focuses on analyzing the geometric characteristics of different embedding spaces. We propose considering the incorporation of the geometric notions discussed in this work into new scalable architectures, such as the NodeFormer, as part of future research.

# 7 ACKNOWLEDGMENT AND FUNDING

AK acknowledges financial support from the NSERC Discovery Grant No. RGPIN-2023-04482 and their McMaster Startup Funds.

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

## A    ADDITIONAL BACKGROUND

Many of the proofs or our paper's results, and generalizations thereof contained only in the manuscript's appendix, rest on some additional technology from the theory of metric spaces. This brief appendix overviews those tools. We also include formal definitions of the involved MLPs with $\mathrm{ReLU}$ activation function which we routinely use.

### A.1    METRIC SPACES

Suppose that $(X, d)$ is a metric space; i.e. $C = 1$ in (iii) above. Topologically, $(X, d)$ and its snowflakes $(X, d^p)$ are identical but geometrically they are quite different. Geometrically, the latter much more complex than the former. In this paper we quantify complexity, when required, using the *doubling dimension* and *aspect ratio* of a metric space.

Denote a ball of radius $r \geq 0$ in $(X, d)$ about a point $x \in X$ is the set $B(x, r) \stackrel{\text{def.}}{=} \{u \in X : d(x, u) < r\}$. The *doubling dimension* of $(X, d)$, denoted by $\dim(X, d)$, is the smallest integer $k$ for which: for every ball $B(x, r)$ about any point $x \in X$ and radius $r > 0$ there are $x_1, \dots, x_{2^k} \in X$ covering it by balls of half its radius; i.e. the metric ball $B(x, r) \subseteq \bigcup_{i=1}^{2^k} B(x_i, r/2)$. We emphasize

that metric tools are finer then topological tools, for instance the dimension of many metric spaces[4] Le Donne & Rajala (2015) is often by much greater than their topological dimension.

We make use of the *aspect ratio* of a weighted graph $\mathcal{G} = (V, E, W)$, as the ratio of the largest distance $d_\mathcal{G}$ between any two nodes over the smallest possible distance between any two nodes, with respect to the geodesic distance on $\mathcal{G}$. This aspect ratio coincides with the aspect ratio used in Kratsios et al. (2023a) and a metric variant of the aspect ratio of the measure theoretic aspect ratio of Krauthgamer et al. (2005). Briefly, when $\#V > 1$ we define

$$\text{aspect}(\mathcal{G}) \stackrel{\text{def.}}{=} \frac{\max_{v,u \in V} d_\mathcal{G}(u,v)}{\min_{v,u \in V; \, u \neq v} d_\mathcal{G}(u,v)},$$

otherwise, we set $\text{aspect}(\mathcal{G}) = 1$. In the case where $V \subset \mathbb{R}^D$, for some $D \in \mathbb{N}_+$, and $W(u,v) = \|u - v\|$ we write $\text{aspect}_2(V) \stackrel{\text{def.}}{=} \text{aspect}(\mathcal{G})$.

We often reply on the notion of a *bi-Lipschitz* embedding, which we now recall. Given any $0 < s \leq L < \infty$, metric spaces $(X, d)$ and $(Y, \rho)$, and a map $f : \mathcal{X} \to \mathcal{Y}$ is called $(s, L)$-*bi-Lipschitz* if

$$s \, d(u,v) \leq \rho(f(u), f(v)) \leq L \, d(u,v) \tag{4}$$

for each $u, v \in \mathcal{X}$. If, $s = L = 1$, then the map $f$ is said to be an isometric embedding[5].

## A.2 MLPs with ReLU Activation Function

We will often be using **M**ulti-**l**ayer **P**erceptron (MLPs) based on the perceptron model of Rosenblatt (1958), and typically called feedforward neural networks in the contemporary approximation theory literature Mhaskar & Poggio (2016); Yarotsky (2017); Petersen & Voigtlaender (2018); Bolcskei et al. (2019); Elbrächter et al. (2021); Galimberti et al. (2022); Daubechies et al. (2022); Marcati et al. (2022); Adcock et al. (2020); Shen et al. (2022). Our MLPs will always use the $\text{ReLU}(t) \stackrel{\text{def.}}{=} \max\{0, t\}$ activation function, where $t \in \mathbb{R}$. We will routinely apply the ReLU function *component-wise*, any vector $u \in \mathbb{R}^d$ for any positive integer $N$, denoted by $\text{ReLU} \bullet u$ and defined by

$$\sigma \bullet u \stackrel{\text{def.}}{=} (\sigma(u_i))_{i=1}^N. \tag{5}$$

For any pair of positive integers $D, d \in \mathbb{N}_+$, a map $f : \mathbb{R}^D \to \mathbb{R}^d$ is called an *MLP* if it admits the recursive representation

$$\begin{aligned}
f(x) &\stackrel{\text{def.}}{=} A^{(J)} x^{(J)} + b^{(J)}, \\
x^{(j+1)} &\stackrel{\text{def.}}{=} \sigma \bullet (A^{(j)} x^{(j)} + b^{(j)}) \quad \text{for } j = 0, \dots, J-1, \\
x^{(0)} &\stackrel{\text{def.}}{=} A^{(0)} x.
\end{aligned} \tag{6}$$

for positive integers $d_0, \dots, d_J, d_{J+1}, J$ with $d_0 = d$ and $d_{J+1} = D$, $d_j \times d_{j+1}$-matrices $A^{(j)}$, and vectors $b^{(j)} \in \mathbb{R}^{d_{j+1}}$. The depth of (the representation equation 6 of) $f$ is $D + 1$, the number of *hidden layers* of (the representation equation 6 of) $f$ is $D$, the width $W(f)$ of (the representation equation 6 of) $f$ is $\max_{j=0,\dots,J+1} d_j$, and the number of trainable/non-zero parameters $P(f)$ of (the representation equation 6 of) $f$ is

$$P(f) \stackrel{\text{def.}}{=} \sum_{j=0}^{J+1} \|A^{(j)}\|_0 + \|b^{(j)}\|_0 \leq W(f)^2 (D + 1)$$

where $\|A^{(j)}\|_0$ (resp. $\|b^{(j)}\|_0$) counts the number of non-zero entries of $A^{(j)}$ (resp. of $b^{(j)}$). We remark that the estimate $P(f) \leq W(f)^2 (D + 1)$ is often larger for several types neural networks architectures, e.g. convolutional neural networks with downsampling layers Zhou (2020a;b).

## B Detailed Model Complexities

This appendix contains a detailed version of Table 1 with fully explicit constants.

---

[4]This is true for several sub-Riemannian manifolds, for instance
[5]Some authors call the special case where $s = \frac{1}{L}$ an $L$-quasi-isometry.

*Table 5:* Details Complexity of the Neural Snowflakes and MLPs.

| Geometry | Net. | Hidden Layers (Depth $-1$) | Width | Theorem |
|---|---|---|---|---|
| General | $f$ | 1 | 3 | 5 |
| Bounded | $f$ | 1 | $\lceil \frac{I(I-1)+2}{4} \rceil$ | 4 |
| All | $\mathcal{E}$ | $\mathcal{O}\left( I \left\{ 1 + \sqrt{I \log(I)} \left[ 1 + \frac{\log(2)}{\log(I)} \left( C_D + \frac{\log\left( I^2 \ \mathrm{aspect}_2(V) \right)}{\log(2)} \right)_+ \right] \right\} \right)$ | $D(I-1) + \max\{d, 12\}$ | 4 and 5 |

$I = \#V$ and the "dimensional constant" is $C_D \stackrel{\text{def.}}{=} \frac{2\log(5\sqrt{2\pi}) + \frac{3}{2}\log(D) - \frac{1}{2}\log(D+1)}{2\log(2)} > 0.$

## C  PROOFS

This section contains derivations of our paper's main results.

### C.1  LEMMATA

We can generate functions satisfying the conditions of Proposition 1 using the following lemma.

**Lemma 1** (Pinelis)**.** *A function* $f : [0, \infty) \to [0, \infty)$ *is continuous, concave, and monotonically increasing if and only if there is a non-negative decreasing function* $g : \mathbb{R} \to \mathbb{R}$ *satisfying*

$$f(t) = \int_0^t g(s)\, ds$$

*for each* $t \in [0, \infty)$.

**Example 4.** *For example, for any* $0 < a \le 1$ *and* $0 \le b \le 1 - a$, *the map* $f : [0, \infty) \to [0, \infty)$ *given by* $f(t) \stackrel{\text{def.}}{=} t^a \ln(1+t)^b$ *is continuous, concave, and monotonically increasing since*

$$f(t) = \int_0^t f(s) \left( \frac{a}{s} + \frac{b}{(1+s)\log(1+s)} \right) ds$$

*and* $t \mapsto f(t) \left( \frac{a}{t} + \frac{b}{(1+t)\log(1+s)} \right)$ *is a non-negative decreasing function on* $\mathbb{R}$.

The following helpful lemma in constructing functions satisfying the conditions of Proposition 1 from more elementary ones.

**Lemma 2** ((Boyd & Vandenberghe, 2004, page 102))**.** *Let* $C \subseteq \mathbb{R}^d$ *be a non-empty convex set and* $d \in \mathbb{N}_+$. *If* $f : C \to [0, \infty)$ *is concave, and* $g : [0, \infty) \to [0, \infty)$ *is non-decreasing and concave, then* $g \circ f : C \to \mathbb{R}$ *is concave. If* $C = [0, \infty)$ *and if* $f$ *and* $g$ *are increasing, then so is* $g \circ f$.

**Lemma 3** (Exponential Transformations)**.** *Let* $a > 0$. *The real-valued map* $f$ *on* $[0, \infty)$ *given for each* $t \ge 0$ *by* $f(t) = 1 - e^{-at}$ *satisfies the conditions of Proposition 1.*

*Proof of Lemma 3.* Since $\partial_t f(t) = a\, e^{-at} \ge a > 0$ and $\partial_t^2 f(t) = -a^2\, e^{-at}$, for every $t \in [0, \infty)$, then $f$ is strictly increasing and concave on $[0, \infty)$. Noting that $f(0) = 1 - e^0 = 0$ completes the proof. $\square$

Lemma 3 directly implies that conical combinations of functions satisfying the conditions of Proposition 1 also satisfy the condition of Proposition 1.

**Lemma 4.** *If* $N \in \mathbb{N}_+$, $A \in (0, \infty)^N$, *and* $f^{(1)}, \dots, f^{(N)}$ *satisfy the conditions of Proposition 1, then* $f(t) \stackrel{\text{def.}}{=} \sum_{n=1}^N A_n f^{(n)}(t)$ *satisfies the conditions of Proposition 1.*

**Lemma 5** (Classical Snowflaking Functions)**.** *Let* $0 < \alpha \le 1$. *The real-valued map* $f$ *on* $[0, \infty)$ *given for each* $t \ge 0$ *by* $f(t) = t^\alpha$ *satisfies the conditions of Proposition 1.*

*Proof of Lemma 5.* The case where $\alpha$ is clear; therefore, suppose that $\alpha < 1$. For every $t \in (0, \infty)$, we have that $\partial_t f(t) = \alpha t^{\alpha-1} > 0$ and $\partial_t^2 f(t) = (1-\alpha)\alpha t^{\alpha-2} = (1-\alpha^2)t^{\alpha-2} < (1-\alpha)t^{\alpha-2} < 0$. Therefore, $f$ is strictly increasing and concave on $[0, \infty)$. Noting that $f(0) = 0^\alpha = 0$ completes the proof. $\square$

**Lemma 6** (Logarithmic Snowflaking Functions). *Let $0 < \beta \leq 1$. The real-valued map $f$ on $[0, \infty)$ given for each $t \geq 0$ by $f(t) = \log(1 + |t|)^\beta$ satisfies the conditions of Proposition 1.*

*Proof of Lemma 6.* Suppose that $\beta = 1$. Since $\log(1 + |t|) = \log(1) = 0$ then $f(0) = 0$. For every $t \in (0, \infty)$, we have that $\partial_t f(t) = \frac{1}{1+t} 0$ and $\partial_t^2 f(t) = -\frac{1}{1+t^2} < 0$. Therefore, $f$ is strictly increasing and concave on $[0, \infty)$. By Lemma 3 $t \mapsto t^\beta$ is increasing and concave on $(0, \infty)$ and since $f$ was strictly increasing and concave, then Lemma 2 yields the conclusion. $\square$

### C.2 PROOF OF THEOREM 1

*Proof of Theorem 1.*
*Step 1 - Characterization of Edges by Cardinality of Geodesic Unit Balls*
Let $n \stackrel{\text{def.}}{=} \#V$. Fix any enumeration $V = \{v_i\}_{i=1}^n$. For each $i \in \{1, \ldots, n\}$ set

$$k_i \stackrel{\text{def.}}{=} \#\{w \in V : \{u_i, w\} \in E\}.$$

Note that, by definition, $k_i \in \{1, \ldots, n\}$. Since $G$ is unweighted, then $W(\{u_i, u_j\}) = 1$ for each $i, j \in [n]$. Consequentially, for each $i, j \in [n]$ if $\{u_i, u_j\} \in E$ then $(u_i = x_1, x_2 = u_j)$ is a minimal path of length one, from $u_i$ to $u_j$. Therefore, for each $i, j \in [n]$ we have that

$$d_G(u_i, u_j) = \inf_{(x_1, \ldots, x_k)} \sum_{i=1}^{k-1} W(\{x_i, x_{i+1}\}) \tag{7}$$

$$= \inf_{(x_1, \ldots, x_k)} \sum_{i=1}^{k-1} 1$$

$$= \inf_{(x_1, \ldots, x_k)} (k - 1)$$

$$= 1 \tag{8}$$

where the infimum is taken over all paths $(u_i = x_1, \ldots, x_k = u_j)$ on $G$ from $u_i$ to $u_j$. Thus, equation 7-equation 8 imply that: for each $i \in [n]$

$$\#\{w \in V : d_G(u_i, w) = 1\} = \#\{w \in V : \{u_i, w\} \in E\} = k_i. \tag{9}$$

By construction, for each $i \in [n]$, $r(k_i) = 1$ and there exists exactly $k_i$ elements in $\overline{B}_{(V,d_G)}(u_i, r(k_i)) \stackrel{\text{def.}}{=} \{v \in V : d_G(u_i, v) \leq r(k_i)\}$. Consequentially,

$$\{u_i, u_j\} \in E \qquad \Leftrightarrow \qquad j \leq i^\star \text{ and } d_G(u_i, u_j) \leq 1 = r(k_i). \tag{10}$$

In particular, $i^\star = \#(\overline{B}_{(V,d_G)}(u_i, r(k_i)) \cap V\}$; meaning that the condition $j \leq i^\star$ can be dropped. Thus, equation 10 simplifies to: for each $i \in [n]$

$$j \leq i^\star \text{ and } d_G(u_i, u_j) \leq r(k_i) \qquad \Leftrightarrow \qquad d_G(u_i, u_j) \leq r(k_i) \tag{11}$$

Incorporating equation 11 into equation 10 allows us to further simplify to: for all $i, j \in [n]$

$$\{u_i, u_j\} \in E \qquad \Leftrightarrow \qquad d_G(u_i, u_j) \leq r(k_i) \tag{12}$$

*Step 2 - Reformulation via Embeddings*
Since the graph inference model $(\mathfrak{E}, \mathfrak{R})$ is universal, in the sense of Definition 1, then there exists a quasi-metric space $(\mathcal{R}, d_\mathcal{R}) \in \mathfrak{R}$ and an encoder $\mathcal{E} : \mathbb{R}^D \to \mathcal{R}$ in $\mathfrak{E}$ such that for each $i, j \in [n]$

$$d_\mathcal{R}(\mathcal{E}(u_i), \mathcal{E}(u_j)) = d_G(u_i, u_j). \tag{13}$$

Combining equation 13 and equation 12 implies that: for each $i, j \in [n]$ the pair $\{u_i, u_j\}$ belongs to $E$ if and only if

$$d_\mathcal{R}(\mathcal{E}(u_i), \mathcal{E}(u_j)) = d_G(u_i, u_j) \leq 1 = r(k_i). \tag{14}$$

Combining equation 14 with equation 11 yields: for each $i, j \in [n]$

$$\{u_i, u_j\} \in E \qquad \Leftrightarrow \qquad j \leq i^\star \text{ and } d_\mathcal{R}(\mathcal{E}(u_i), \mathcal{E}(u_j)) \leq r(k_i).$$

This concludes the proof. $\square$

*Proof of Theorem 2.* **Constructing The Isometric Embedding into a Euclidean Snowflake** Since $(V, d_{\mathcal{G}})$ is an $I$-point metric space. If $I = 1$, there is nothing to show; suppose, therefore, that $I > 1$. Then, (Deza & Maehara, 1990, Corollary 3) implies that for $\varepsilon^{\star} \stackrel{\text{def.}}{=} \log_2(1 + \frac{1}{I-1})/2$ there exists some $d \in \mathbb{N}_+$ and a map $\tilde{\varphi}_{\varepsilon^{\star}} : V \to \mathbb{R}^{d_{\varepsilon^{\star}}}$ satisfying

$$d_{\mathcal{G}}(u, v)^{\varepsilon^{\star}} = \|\varphi(v) - \varphi(u)\|. \tag{15}$$

As shown in Schoenberg (1937), equation 15 implies that the statement holds (mutatis mondanis) for any other $\varepsilon \in (0, \varepsilon^{\star}]$ for some other map $\tilde{\varphi}_{\varepsilon} : V \to \mathbb{R}^{d_{\varepsilon}}$ into some Euclidean space $\mathbb{R}^{d_{\varepsilon}}$. Fix any such $\varepsilon$, set $p \stackrel{\text{def.}}{=} 1/\varepsilon$ and let $\varphi$ by any extension of $\tilde{\varphi}_{\varepsilon}$ defined on all of $\mathbb{R}^D$.

(Kratsios et al., 2023a, Lemma 20) implies that is an MLP with ReLU activation function $E : \mathbb{R}^D \to \mathbb{R}^d$ satisfying $\mathcal{E}(u) = \varphi(u)$ for all $u \in V$. Whence, equation 15 implies that

$$d_{\mathcal{G}}(u, v) = \|\mathcal{E}(v) - \mathcal{E}(u)\|^p.$$

NB, in the special case where $\mathcal{G}$ is a tree (Maehara, 1986, Theorem 6) we may instead set $\varepsilon^{\star} = 1/2$ and therefore $p$ may be instead taken to be $p = 4$ in the above argument. **Implementing The Snowflaking Function** We now implement the snowflaking map $t \mapsto t^p$ using a neural snowflake $f$; i.e. with representation equation 3. Set $I = 1$, let $p = 2/\big(\log_2(1 + \frac{1}{I-1})\big)$, and consider the parameters

$$A^{(1)} = (1),\ B^{(1)} = (1),\ C^{(1)} = \begin{pmatrix} 0 \\ 1 \\ 0 \end{pmatrix}\ \alpha = 1,\ \beta = 0.$$

Then, $f$ defined by equation 3 satisfies $f(t) = t^p$. Furthermore, $f$ has 1 hidden layer, width 3, and 4 trainable parameters and equation 15 implies that

$$d_{\mathcal{G}}(u, v) = f\big(\|\mathcal{E}(v) - \mathcal{E}(u)\|\big).$$

Since $p = 2/\big(\log_2(1 + \frac{1}{I-1})\big)$ then Example 1 show that $f(\| \cdot - \cdot \|)$ is a quasi-metric space with $2^{2/\big(\log_2(1+\frac{1}{I-1})\big)-1}$-relaxed triangle inequality.

In the special case where $\mathcal{G}$ is a tree, we have $p = 4$. Thus, the neural snowflake supports an 8-relaxed triangle inequality. □

*Proof of Theorem 5.* **Existence and Memorization of the Snowflake Embedding** Set $\varepsilon \stackrel{\text{def.}}{=} 1 - p^{-1}$ and note that $\varepsilon \in (1/2, 1)$. By (Naor & Neiman, 2012, Theorem 2), there exists $D, N > 0$ and a map $\varphi : (V, d_G^{1-\varepsilon}) \to \mathbb{R}^N$ satisfying: for every $x, u \in V$

$$d_G(x, u)^{1-\varepsilon} \le f\big(\|\varphi(x) - \varphi(u)\|\big) \le D\, d_G(x, u)^{1-\varepsilon}, \tag{16}$$

where $N = c_1 \log(\dim(G))$ and $D = c_2 \frac{\log(\dim(G))^2}{\varepsilon^2}$, for absolute constants $c_1, c_2 > 0$ independent of $G$ and of $p$. Set $f(t) \stackrel{\text{def.}}{=} t^p = t^{1/(1-\varepsilon)}$. Since $f$ is monotone increasing, then applying it through the inequalities in equation 16 yields

$$d_G(x, u) \le \|\varphi(x) - \varphi(u)\| \le D^p\, d_G(x, u), \tag{17}$$

for every $x, u \in V$.

By (Kratsios et al., 2023a, Lemma 20), there exists an MLP $\mathcal{E} : \mathbb{R}^d \to \mathbb{R}^D$ with ReLU activation function such that, for every $x \in V$ we have $\varphi(x) = \mathcal{E}(x)$. Therefore, equation 17 implies that

$$d_G(x, u) \le \|\varphi(x) - \varphi(u)\| \le D^p\, d_G(x, u), \tag{18}$$

for each $u, u \in V$. Furthermore, the depth, width, and number of trainable parameters defining $\mathcal{E}$ are as in (Kratsios et al., 2023a, Lemma 20) and a recorded in Table 5 (with abbreviated versions recorded in Table 1).

**Implementing The Snowflaking Function** As in the proof of Theorem 2, we now implement the snowflaking map $t \mapsto t^p$ using a neural snowflake $f$; i.e. with representation equation 3. Set $I = 1$, let $p$ in equation 3 be the as in equation 17, and consider the parameters

$$A^{(1)} = (1),\ B^{(1)} = (1),\ C^{(1)} = \begin{pmatrix} 0 \\ 1 \\ 0 \end{pmatrix}\ \alpha = 1,\ \beta = 0.$$

Then, $f$ defined by equation 3 satisfies $f(t) = t^p$. Furthermore, $f$ has 1 hidden layer, width 3, and 4 trainable parameters. □

### C.3 BOUNDED COMPONENT

For the proof of the next result, we recall that a function $f : [0, \infty) \to [0, \infty)$ is said to be *completely monotone* if $f$ is continuous on $[0, \infty)$, smooth on $(0, \infty)$, and its derivatives satisfy the following alternating sum property

$$(-1)^n \partial_t^n f(t) \geq 0 \tag{19}$$

for every $t > 0$ and every $n \in \mathbb{N}_+$. See (Widder, 1941, Chapter IV) for a detailed study of completely monotone functions and several examples thereof.

In particular, this and Proposition 1 imply that $-f$ is monotonically increasing and concave, therefore the map, $\bar{f}$, defined for $t \geq 0$ by $\bar{f}(t) \stackrel{\text{def.}}{=} f(0) - f(t)$ produces a well-defined snowflake metric $d_{\bar{f}} \stackrel{\text{def.}}{=} \bar{f}(\| \cdot - \cdot \|)$. Since $f$ is completely monotone then it is monotonically decreasign on $[0, \infty)$ and bounded below by 0; whence, $\bar{f}(t)$ is monotonically increasing and contained in $[0, f(0)]$. Therefore, $d_{\bar{f}}$ is *bounded* between $[0, f(0)]$. We will show that the neural snowflake can generate a snowflake metric which interpolates any bi-Lipschitz embedding into such a space.

Before proving Theorem 4 we note that it holds under the following alternative assumption to Assumption 1. Intuitively, this assumptions state that the map $f$, in Assumption 1 can be taken to be the moment-generating function (MGF) of some probability measure on $[0, \infty)$.

**Assumption 2** (Alternative to Assumption 1: Latent MGD Geometry). *There are $d, D \in \mathbb{N}_+$, a latent feature map $\Phi : \mathbb{R}^D \to \mathbb{R}^d$, and a Borel probability measure $\mathbb{P}$ on $[0, \infty)$ whose MGF $f(t) = \mathbb{E}_{X \sim \mathbb{P}}[e^{-t X}]$ exists for all $t \geq 0$, is non-constant, and satisfies*

$$d_{\mathcal{G}}(x, y) = \bar{f}\big(\|\Phi(x) - \Phi(y)\|\big),$$

*where $\bar{f}(t) \stackrel{\text{def.}}{=} f(0) - f(t)$.*

Both Assumptions 1 and 2 are special cases of the following more general assumption.

**Assumption 3** (Kernel or Moment-Generating Priors). *Suppose that $f : [0, \infty) \to [0, \infty)$ is non-constant and either:*

*(i) $f(t) = \int_0^\infty e^{-t u} \mu(du)$ for some finite Borel measure $\mu$ on $[0, \infty)$,*

*(ii) For each $k \in \mathbb{N}_+$, the map $K_f : (x, y) \in \mathbb{R}^k \times \mathbb{R}^k \mapsto f(x^\top y) \in \mathbb{R}$ is a positive-definite kernel on $\mathbb{R}^k$.*

*Define $\bar{f}(t) \stackrel{\text{def.}}{=} f(0) - f(t)$.*

The following result implies, and generalizes, Theorem 4, to bi-Lipschitz embeddings.

**Proposition 3** (Embeddings into Bounded Metric Spaces - Bi-Lipschitz Version). *Let $D, d \in \mathbb{N}_+$, $f$ satisfy Assumption 3, and $G$ be a finite weighted graph with $V \subset \mathbb{R}^d$. Then, $\| \cdot - \cdot \|_{\bar{f}}$ defines a bounded metric on $\mathbb{R}^d$ and for every $(s, L)$-bi-Lipschitz embedding $\Phi : (V, d_G) \to (\mathbb{R}^d, \|\cdot\|_{\bar{f}})$ there is a neural snowflake $(\mathcal{E}, f)$ satisfying*

$$\|\Phi(v) - \Phi(u)\|_f = \|\mathcal{E}(v) - \mathcal{E}(u)\|_{\bar{f}}$$

*for every $u, v \in V$. In particular, for each $u, v \in V$ we have*

$$s \, d_G(v, u) \leq \|\mathcal{E}(v) - \mathcal{E}(u)\|_f \leq s \, L \, d_G(v, u).$$

*Furthermore, the depth, width, and number of trainable parameters of $\mathcal{E}$ and of $f$ are as in Table 1.*

*Proof of Theorem 3.* **Rephrasing as completely monotone functions** Suppose that Assumption 3 holds. The Hausdorff-Bernstein-Widder theorem, see (Widder, 1941, Theorem IV.12a), implies that $f$ is completely monotone. Alternatively, suppose that Assumption 3 (ii) holds then Schoenberg's theorem, see[6] (Schoenberg, 1938, Theorem 3), implies that $f$ is completely monotone.

Since $f$ is defined on all of $[0, \infty)$ then equation 19 implies that $f(t) \geq 0$ for all $t$ whence takes non-negative valued. Likewise equation 19 implies that $\partial_t f(t) \leq 0$ therefore $f$ is non-increasing;

---

[6]See Phillips et al. (2019) for more general version.

whence $\bar{f}$ is bounded in $[0, f(0)]$. Finally, equation 19 implies that $\partial_t^2 f(t) \geq 0$ therefore $f$ is convex; thus $\bar{f}$ is concave since $\partial_t^2 \bar{f} \leq 0$. Whence Proposition 1 implies that $\| \cdot - \cdot \|_{\bar{f}}$ is a metric on $\mathbb{R}^D$.

**Interpolating $\mathcal{E}$ and $\bar{f}$**

Enumerate $V = \{x_i\}_{i=1}^I$ where $I \overset{\text{def.}}{=} \#I$. Consider an $(s, L)$-bi-Lipschitz embedding $\Phi : (V, d_G) \to (\mathbb{R}^d, \| \cdot \|_f)$. By (Kratsios et al., 2023a, Lemma 20), there exists an MLP $\mathcal{E} : \mathbb{R}^D \to \mathbb{R}^d$ with ReLU activation function satisfying

$$\Phi(v) = \mathcal{E}(v) \tag{20}$$

for each $v \in V$. Moreover, the depth, width, and number of trainable parameters determining $E$ are as in (Kratsios et al., 2023a, Lemma 20) and are recorded in Table 5 (and abbreviated in Table 1).

Since $\Phi$ is bi-Lipschitz then it is injective; whence $\{\|\Phi(x_i) - \Phi(x_j)\|\}_{i,j=1}^I$ has exactly as many points as $\{\|x_i - x_j\|\}_{i,j=1}^I$. By symmetry of the Euclidean metric, observe that the number $\tilde{I}$ of elements in set $\{\|\Phi(x_i) - \Phi(x_j)\|\}_{i,j=1}^I \cup \{0\}$ is at-most $I(I - 1)/2 + 1$ elements; which we sort and enumerate $\{\|x_i - x_j\|\}_{i,j=1}^I \cup \{0\}$ by $\{t_i\}_{i=1}^{\tilde{I}}$. By the (McGlinn, 1978, Corollary on page 215) there exists a unique exponential sum $Y(t) = \sum_{i=1}^{\lceil \tilde{I}/2 \rceil} \beta_i\, e^{-\alpha_i t}$ satisfying

$$Y(t_i) = f(t_i) \tag{21}$$

for every $i = 0, \ldots, \tilde{I}$, and in particular $Y(0) = f(0)$ since $0 \in \{t_i\}_{i=1}^{\tilde{I}}$. Since $\bar{f}(t) = f(0) - f(t)$ for all $t \geq 0$, then equation 21 implies that

$$
\begin{aligned}
\bar{f}(t_i) &= f(0) - f(t_i) \tag{22}\\
&= Y(0) - Y(t_i)\\
&= \left(\sum_{i=1}^{\lceil \tilde{I}/2 \rceil} \beta_i\, e^{-\alpha_i 0}\right) - \left(\sum_{i=1}^{\lceil \tilde{I}/2 \rceil} \beta_i\, e^{-\alpha_i t_i}\right)\\
&= \sum_{i=1}^{\lceil \tilde{I}/2 \rceil} \left(\beta_i\, e^{-\alpha_i 0} - \beta_i\, e^{-\alpha_i t_i}\right)\\
&= \sum_{i=1}^{\lceil \tilde{I}/2 \rceil} \left(\beta_i\, 1 - \beta_i\, e^{-\alpha_i t_i}\right)\\
&= \sum_{i=1}^{\lceil \tilde{I}/2 \rceil} \beta_i\left(1 - e^{-\alpha_i t_i}\right) \overset{\text{def.}}{=} \bar{Y}(t_i), \tag{23}
\end{aligned}
$$

for all $i = 1, \ldots, \tilde{I}$. In particular, equation 22-equation 23, the definition of the $t_i$, and the memorization/interpolation guarantee in equation 20 implies that

$$f\big(\|\Phi(x_i) - \Phi(x_j)\|\big) = \bar{Y}\big(\|\mathcal{E}(x_i) - \mathcal{E}(x_j)\|\big) = \bar{Y}\big(\|\mathcal{E}(x_i) - \mathcal{E}(x_j)\|\big) \tag{24}$$

for $i, j = 1, \ldots, I$ (note $0 = \|\mathcal{E}(x_1) - \mathcal{E}(x_1)\|$ so $f(0) = Y(0)$ is implied by equation 24). Therefore, for each $i, j = 1, \ldots, I$ we have

$$
\begin{aligned}
\|\mathcal{E}(x_i) - \mathcal{E}(x_j)\|_{\bar{f}} &= \|\Phi(x_i) - \Phi(x_j)\|_{\bar{f}}\\
&\overset{\text{def.}}{=} \bar{f}\big(\|\Phi(x_i) - \Phi(x_j)\|\big)\\
&= \bar{Y}\big(\|\Phi(x_i) - \Phi(x_j)\|\big)\\
&= \|\Phi(x_i) - \Phi(x_j)\|_{\bar{Y}}.
\end{aligned}
$$

It remains to show that $\bar{Y}$ can be implemented by a map, which we denote by $\tilde{f}$, with representation equation 3.

**Implementing The Exponential Sum** In the notation of equation 3, set $I = 1$ and consider

$$A^{(1)} = \begin{pmatrix} \alpha_1 \\ \vdots \\ \alpha_M \end{pmatrix},\ B^{(1)} = (\beta_1 \ldots \beta_M),\ C^{(1)} = \begin{pmatrix} 1 \\ 0 \\ 0 \end{pmatrix},\ d = (0),\ \alpha_1 = 0 = \beta_0. \tag{25}$$

Then, the map $\tilde{f}$ defined as in equation 3 with parameters equation 25 is

$$\tilde{f}(t) = \left(B^{(1)}\, \sigma_{0,0}(A^{(1)}t)C^{(1)}\right)^{1+|0|} = \sum_{i=1}^{M} \beta_i\,(1 - e^{-\alpha_i\, t}) = \bar{Y}(t)$$

for every $t \geq 0$. Tallying parameters in $\tilde{f}$, shows that $\tilde{f}$ has $3M + 1$ non-zero parameters, 1 hidden layer, and width $M$; as recorded in Table 5 (and abbreviated in Table 1). $\qquad\square$

### C.4   Detailed Euclidean Comparisons

Together, the following Propositions imply Theorem 3.

**Proposition 4** (All Embeddings Implementable MLPs are Implementable by a Neural Snowflakes)**.** *For any $d, D, P \in \mathbb{N}_+$, $0 \leq s \leq L$. For any weighted finite graph $G = (E, V, W)$ with $V \subset \mathbb{R}^D$: if there is an MLP $\tilde{\mathcal{E}} : \mathbb{R}^D \to \mathbb{R}^d$ with $P$ non-zero parameters satisfying*

$$s\, d_G(v, u) \leq \|\tilde{\mathcal{E}}(v) - \tilde{\mathcal{E}}(u)\| \leq L\, d_G(v, u)$$

*for each $v, u \in V$. There is a pair of an MLP $\mathcal{E} : \mathbb{R}^D \to \mathbb{R}^d$ and a neural snowflake $f$ satisfying*

$$s\, d_G(v, u) \leq \|\tilde{\mathcal{E}}(v) - \tilde{\mathcal{E}}(u)\|_f \leq L\, d_G(v, u)$$

*for each $v, u \in V$. Furthermore, the total number of non-zero parameters in $\mathcal{E}$ and $f$ is $P + 4$.*

*Proof of Proposition 4.* Set $\mathcal{E} \stackrel{\text{def.}}{=} \tilde{\mathcal{E}}$; in particular, $\mathcal{E}$ is defined by $P$ parameters.

It remains to show that $f$ can implement the identity function. In the notation of equation 3, set $I = 1$, and consider the parameters

$$A^{(1)} = (1),\ B^{(1)} = (1),\ C^{(1)} = \begin{pmatrix} 0 \\ 1 \\ 0 \end{pmatrix}\ \alpha = 1,\ \beta = 0.$$

Then, $f$ defined by equation 3 satisfies $f(t) = t$. Furthermore, $f$ has 1 hidden layer, width 3, and 4 trainable parameters. The conclusion now follows since $\tilde{\mathcal{E}}$ was assumed to implement an $(s, L)$-bi-Lipschitz embedding of $(V, d_G)$ into $(\mathbb{R}^d, \|\cdot\|)$. $\qquad\square$

**Proposition 5** (Neural Snowflakes can Implement Isometric Embeddings which MLPs Cannot)**.** *For any $d, D \in \mathbb{N}_+$ there exists a fully-connected weighted graph $G = (V, E, W)$ with $V \subset \mathbb{R}^D$ which:*

*(i)  Cannot be isometrically embedded into $(\mathbb{R}^n, \|\cdot\|)$ for any $n \leq d$,*

*(ii)  There exists a neural snowflake $(\mathcal{E}, f)$ with $\mathcal{E} : \mathbb{R}^D \to \mathbb{R}^d$ satisfying: for each $x, u \in V$*
$$\|\mathcal{E}(x) - \mathcal{E}(u)\|_f = d_G(x, u).$$

*Proof of Proposition 5.* Fix $d, D \in \mathbb{N}_+$, $\alpha = 1/2$, and $p \stackrel{\text{def.}}{=} \alpha$. Then, (Le Donne et al., 2018, Theorem 1.1) implies that there exists some $N \in \mathbb{N}_+$ such that for any metric space $(V, d_G)$ with at-least $N$ points, the $1/2$-snowflake $(V, d_G^{1/2})$ cannot be isometrically embedded in $(\mathbb{R}^d, \|\cdot\|)$. If $d > 1$, suppose that for some $n < d$, $(V, d_G^{1/2})$ admitted an isometric embedding $\varphi_1 : (V, d^{1/2}) \to (\mathbb{R}^n, \|\cdot\|)$ then, since the map $\varphi_2 : \mathbb{R}^n \to \mathbb{R}^d$ given by $z \mapsto (z_1, \ldots, z_n, 0, \ldots, 0)$, and since the composition of isometries is again an isometry, then $\varphi \stackrel{\text{def.}}{=} \varphi_2 \circ \varphi_1$ would define an isometry from $(V, d_G^{1/2})$ to $(\mathbb{R}^d, \|\cdot\|)$; which is a contradiction. This, yields (i).

Let us now show (ii). Set $p = 2$ and let $V$ be any finite subset of $\mathbb{R}^D$ with $N$ points. Consider the fully-connected graph $G = (V, E, W)$ with edge weights given by

$$W(x, u) \stackrel{\text{def.}}{=} \|x - u\|^{1/2}.$$

By construction the graph geodesic distance $d_G$ on $G$ satisfies $d_G(x, u) = W(x, u)$ for every $x, u \in V$. Furthermore, again by construction, for every $x, u \in V$ we have

$$d_G(x, u) = W(x, u) = \|x - u\|^{1/2} = \|\mathcal{E}(x) - \mathcal{E}(u)\|_f$$

where $f(t) = |t|^{1/2}$ and $\mathcal{E}(x) = x$. Applying (Kratsios et al., 2023a, Lemma 20), we find that there exists an MLP with ReLU activation function $\Phi : \mathbb{R}^D \to \mathbb{R}^d$ with, depth, width, and number of non-zero parameters specified therein, satisfying $\Phi(x) = (x)$ for every $x \in V$. $\qquad\square$

## C.5 PROOF OF IMPOSSIBILITY RESULTS - PROPOSITION 2

We now prove Proposition 2 by showing that the graph depicted in Figure 1 cannot be embedded isometrically into any Riemannian manifold, as we now show.

*Proof of Proposition 2.* Fix $D \in \mathbb{N}_+$ and let $V \subseteq \mathbb{R}^D$ be any 5-point subset; whose points we list by $V = \{A, B, C, D, E\}$. Define the set of edges

$$E \stackrel{\text{def.}}{=} \big\{\{A, E\}, \{A, D\}, \{E, B\}, \{B, D\}, \{D, C\}\big\}$$

and consider the graph $\mathcal{G} \stackrel{\text{def.}}{=} (V, E)$. It is easy to see that $\mathcal{G}$ is connected; thus, the shortest path (geodesic) distance $d_{\mathcal{G}}$ on $V$ is well-defined. Furthermore, there are unique shortest paths joining $C$ to $A$ and $C$ to $B$; which we denote by $[C : A]$ and $[C : B]$ respectively; these are given by the ordered tuples (ordered pairs in this case)

$$[C : A] \stackrel{\text{def.}}{=} \big(\{C, D\}, \{D, A\}\big) \text{ and } [C : B] \stackrel{\text{def.}}{=} \big(\{C, D\}, \{D, B\}\big). \tag{26}$$

We now argue by contradiction.

Suppose that there exists a complete and connected smooth Riemannian manifold $(\mathcal{R}, g)$ and an isometric embedding $\varphi : (V, d_{\mathcal{G}}) \to (\mathcal{R}, d_g)$; where, $d_g$ denotes the shortest path (geodesic) distance on $(\mathcal{R}, g)$. Since $(\mathcal{R}, g)$ is complete (as a metric space) and connected, then the Hopf-Rinow Theorem (Jost, 2017, Theorem 1.7.1) implies that each pair of points in $\mathcal{R}$ can be joined by a distance minimizing geodesic, i.e. it is geodesically complete (Jost, 2017, Definition 1.7.1). Therefore, there exists a pair of geodesics $\gamma_{[C:A]} : [0, 1] \to \mathcal{R}$ and $\gamma_{[C:B]} : [0, 1] \to \mathcal{R}$ satisfying

$$\gamma_{[C:i]}(0) = \varphi(C) \text{ and } \gamma_{[C:i]}(1) = \varphi(i) \tag{27}$$

for $i \in \{A, B\}$. Since $\varphi$ is an isometric embedding and $\{C, D\} \in [C : A] \cap [C : B]$ then equation 26 implies that there is some $0 < t_2 < 1$ for which

$$\gamma_{[C:A]}(t_2) = \gamma_{[C:B]}(t_2) = \varphi(D). \tag{28}$$

Now, equation 27 and the local uniqueness of geodesics in $(\mathcal{R}, g)$ about any point, in particular about $\varphi(D)$ (see (Jost, 2017, Theorem 1.4.2)) imply that there is some $\varepsilon > 0$ such that

$$\gamma_{[C:A]}(s) = \gamma_{[C:B]}(s), \tag{29}$$

for all $t_2 - \varepsilon \leq s \leq t_2 + \varepsilon$. Now since $A \neq B$ and since $\varphi$ is injective then $\varphi(A) \neq \varphi(B)$. However, equation 27 and equation 29 cannot simultaneously hold for $t_2 < s < t_2 + \varepsilon$; thus we have a contradiction. Consequentially, a pair $\big(\varphi, (\mathcal{R}, d_{\mathcal{R}})\big)$ cannot exist. $\qquad\square$

# D DETAILS ON EXAMPLES

This appendix contains further details, explaining derivations of particular examples within the paper's main text.

**Example - Details 1** (Details for Example 2). *This follows from (Phillips et al., 2019, Proposition 3.2) and the Hausdorff-Bernstein-Widder theorem, see (Widder, 1941, Theorem IV.12a), together states that $f \circ u$ satisfies Assumption 2 if $f$ does and if $u : t \mapsto \frac{t}{a+t}$ is a Bernstein function, i.e. a continuous function $f : [0, \infty) \to [0, \infty)$ which is smooth on $(0, \infty)$ and whose derivatives satisfy $(-1)^k \partial_t^k f(t) \leq 0$ for all $t \geq 0$ and all $k \in \mathbb{N}_+$. A long list of Bernstein functions which can be found in the several tables in (Schilling et al., 2012, Section 16.2). For instance, the map $t \mapsto (1 + \sum_{k=1}^K |t|^{r_k})^{-1}$ is one a Bernstein function, see (Schilling et al., 2012, Corollary 6.3), and $f(t) = |t|^\beta$ satisfies Assumption 2.*

**Example - Details 2** (Details for Example 3). *This holds, by arguing as in the previous example, since $t \mapsto e^{-b\,t}$ satisfies Assumption 1 and since $t \mapsto \frac{-(t-1)}{log(t)}$ is a Bernstein function, by (Schilling et al., 2012, Corollary 6.3).*

# E  DISCRETE EDGE SAMPLING AND THE DISCRETE DIFFERENTIABLE GRAPH MODULE

Most latent graph inference models require generating a discrete graph based on a similarity measure between latent node representations. The discrete Differentiable Graph Module (dDGM) (Kazi et al., 2022) has served as a source of inspiration for numerous studies in the field of latent graph inference (Sáez de Ocáriz Borde et al., 2023c;b; Battiloro et al., 2023). It generates a sparse $k$-degree graph using the Gumbel Top-k trick (Kool et al., 2019), a stochastic relaxation of the kNN rule, to sample edges from the probability matrix $\mathbf{P}^{(l)}(\mathbf{X}^{(l)}; \mathbf{\Theta}^{(l)}, T)$, where each entry corresponds to

$$p_{ij}^{(l)} = \exp(-\varphi(\hat{\mathbf{x}}_i, \hat{\mathbf{x}}_j, T)). \tag{30}$$

where $T$ is a learnable temperature parameter, $\hat{x}$ are latent node feature representation, and $\varphi$ is some similarity measure. In practice, the main similarity measure used in Kazi et al. (2022) was to compute the distance based on the features of two nodes in the graph embedding space, which was assumed to be Euclidean. Based on

$$\text{argsort}(\log(\mathbf{p}_i^{(l)}) - \log(-\log(\mathbf{q}))) \tag{31}$$

where $\mathbf{q} \in \mathbb{R}^N$ is uniform i.i.d in the interval $[0, 1]$, we can sample the edges

$$\mathcal{E}^{(l)}(\mathbf{X}^{(l)}; \mathbf{\Theta}^{(l)}, T, k) = \{(i, j_{i,1}), (i, j_{i,2}), ..., (i, j_{i,k}) : i = 1, ..., N\}, \tag{32}$$

where $k$ is the number of sampled connections using the Gumbel Top-k trick. This sampling approach follows the categorical distribution $\frac{p_{ij}^{(l)}}{\Sigma_r p_{ir}^{(l)}}$ and $\mathcal{E}(\mathbf{X}^{(l)}; \mathbf{\Theta}^{(l)}, T, k)$ is represented by the unweighted adjacency matrix $\mathbf{A}^{(l)}(\mathbf{X}^{(l)}; \mathbf{\Theta}^{(l)}, T, k)$. Note that including noise in the edge sampling approach will result in the generation of some random edges in the latent graphs which can be understood as a form of regularization.

# F  GRAPH LEARNING ALGORITHMS: TRAINING AND BACKPROPAGATION

The optimization of the baseline node feature learning component in the architecture, that is, the standard GNN part relies on the loss of the downstream task. In particular, for classification, the cross-entropy loss is utilized. However, it is also necessary to update the parameters of graph learning modules such as the DGM (Kazi et al., 2022), the DCM (Battiloro et al., 2023), and the neural snowflake. To accomplish this, we implement a compound loss that provides incentives for edges contributing to accurate classification while penalizing edges that lead to misclassification. We introduce a reward function

$$\delta(y_i, \hat{y}_i) = \mathbb{E}(ac_i) - ac_i.$$

The aforementioned disparity is calculated as the difference between the mean accuracy of the $i$th sample and the present accuracy of the prediction. Here, $y_i$ and $\hat{y}_i$ represent the true and predicted labels, respectively, while $ac_i$ is assigned a value of 1 if $y_i = \hat{y}_i$, and 0 otherwise. The loss function for graph learning is formulated in terms of the reward function:

$$L_{GL} = \sum_{i=1}^{N} \left( \delta(y_i, \hat{y}_i) \sum_{l=1}^{l=L} \sum_{j:(i,j)\in\hat{\varphi}^{(l)}} \log p_{ij}^{(l)} \right), \tag{33}$$

and it approximates the gradient of the expectation $\mathbb{E}_{(\mathcal{G}^{(1)},...,\mathcal{G}^{(L)})\sim(\mathbf{P}^{(1)},..,\mathbf{P}^{(L)})} \sum_{i=1}^{N} \delta(y_i, \hat{y}_i)$. The expectation $\mathbb{E}(ac_i)^{(t)}$ is calculated based on

$$\mathbb{E}(ac_i)^{(t)} = \beta\mathbb{E}(ac_i)^{(t-1)} + (1-\beta)ac_i, \tag{34}$$

with $\beta = 0.9$ and $\mathbb{E}(ac_i)^{(t=0)} = 0.5$. For further details refer to Kazi et al. (2022) and Sáez de Ocáriz Borde et al. (2023c).

## G  NEURAL SNOWFLAKE FOR LATENT GRAPH INFERENCE ALGORITHMS

This appendix includes Algorithm 1, 2, and 3. They summarize how learned latent graphs are incorporated into a standard GNN pipeline, how the dDGM samples graph edges, and how the neural snowflake architecture computes similarities between latent node representations, respectively. Superscripts are employed to denote layer-specific quantities, while subscripts are utilized for indices.

---

**Algorithm 1:** Node Level Prediction leveraging Inferred Latent Graph (Forward Pass)

---

**Require: $\mathbf{X}, \mathbf{A}$**        ▷ Node Features and Adjacency Matrix
  **return Y**        ▷ Predicted Node Labels
  $\mathbf{X}^{(0)} \leftarrow \mathbf{X}$
  $\hat{\mathbf{A}} \leftarrow \texttt{DGM}(\mathbf{X}^{(0)}, \mathbf{A})$        ▷ Refer to Algorithm 2
  **For** $l = 1$ **to** $L$
    |  $\mathbf{X}^{(l)} \leftarrow \texttt{GNN}^{(l)}(\mathbf{X}^{(l-1)}, \hat{\mathbf{A}})$        ▷ GNN diffusion layers
    **end**
  $\mathbf{Y} \leftarrow \texttt{MLP}(\mathbf{X}^{(L)})$        ▷ Node level prediction

---

Algorithm 2, is a mild modification of the differentiable graph model of Kazi et al. (2022). Briefly, it allows the GNN to update its graph in a differentiable manner.

---

**Algorithm 2:** Discrete Differentiable Graph Module (modified based on Kazi et al. (2022))

---

**Require: $\mathbf{X}, \mathbf{A}$**        ▷ Node Features and Adjacency Matrix
  **return $\hat{\mathbf{A}}$**        ▷ Latent Graph
  $\hat{\mathbf{X}} \leftarrow f(\mathbf{X}, \mathbf{A})$        ▷ Transform node features
  $s_{ij} \leftarrow \texttt{Neural Snowflake}(\hat{\mathbf{x}}_\mathbf{i}, \hat{\mathbf{x}}_\mathbf{j})$        ▷ Compute similarity measures. Refer to Algorithm 3
  $p_{ij} \leftarrow g(s_{ij})$        ▷ Compute edge sampling probabilities based on similarities
  **For** $i = 1$ **to** $N$
    |  $\mathbf{q} \sim U(0,1)$        ▷ Uniform i.i.d.
    |  $\mathbf{j}_{\{k\}} = \texttt{argtopk}(\log\mathbf{p}_i - \log(-\log(\mathbf{q}_i)))$
    |  $\hat{a}_{ij} = \begin{cases} 1 & j \in \mathbf{j}_{\{k\}} \\ 0 & \text{otherwise} \end{cases}$
    **end**
  $\hat{\mathbf{A}} \leftarrow \hat{a}_{ij}$        ▷ Discrete Latent Unweighted Graph Prediction

---

Algorithm 3 describes how the neural snowflake processes node-level features to distances.

## H  COMPUTATIONAL IMPLEMENTATION DETAILS

In this appendix, we present additional information regarding the practical implementation of trainable snowflake activations and neural snowflakes, beyond the theoretical foundation discussed in the main text. We address certain instabilities encountered during the training process and propose methods to mitigate them. Our objective is to provide a deeper understanding of our findings, hoping that this will contribute to the advancement of neural snowflakes in future iterations.

**Hardware and Symbolic Matrices.** In line with previous work, for most of the experiments, we utilized GPUs such as the NVIDIA Tesla T4 Tensor Core with 16 GB of GDDR6 memory, NVIDIA P100 with 16 GB of CoWoS HBM2 memory, or NVIDIA Tesla K80 with 24 GB of GDDR5 memory.

---

**Algorithm 3:** Neural Snowflake Processing

---

**Require:** $\hat{\mathbf{x}}_\mathbf{i}, \hat{\mathbf{x}}_\mathbf{j}$ $\qquad\qquad\qquad\qquad\qquad\qquad\qquad\qquad$ ▷ Two Node Features Vectors
$\quad$ **return** $s_{ij}$ $\qquad\qquad\qquad\qquad\qquad\qquad\qquad\qquad$ ▷ Distance similarity measure
$\quad t \leftarrow ||\hat{\mathbf{x}}_\mathbf{i} - \hat{\mathbf{x}}_\mathbf{j}||_2$ $\qquad\qquad\qquad\qquad\qquad\qquad\qquad\qquad$ ▷ Euclidean Distance
$\quad t^{(0)} \leftarrow t$
$\quad$ **For** $l = 1$ **to** $I$
$\qquad \hat{t}^{(l-1)} \leftarrow A^{(l)} t^{(l-1)}$ $\qquad\qquad\qquad\qquad\qquad\qquad\qquad$ ▷ Linear Projection
$\qquad \Sigma^{(l)} \leftarrow \sigma^{(l)}(\hat{t}^{(l-1)})$ $\qquad\qquad\qquad$ ▷ Trainable Snowflake Activation (Equation 2)
$\qquad t^{(l)} \leftarrow B^{(l)} \Sigma^{(l)} C^{(l)}$ $\qquad\qquad\qquad\qquad\qquad\qquad\qquad$ ▷ Linear Projections
$\qquad$ **end**
$\quad s_{ij} \leftarrow t_I^{1+|p|}$ $\qquad\qquad\qquad\qquad\qquad\qquad\qquad\qquad\qquad$ ▷ Quasi-metric

---

These GPUs have limited memory capacities that are easily surpassed during backpropagation when dealing with datasets other than Cora and CiteSeer. One of the primary computational limitations of the Differentiable Graph Module and other latent graph inference techniques is the necessity to compute distances between latent representations for all nodes in order to generate the latent graph. While the discrete graph sampling method utilized by dDGM offers improved computational efficiency compared to its continuous counterpart, cDGM, due to the creation of sparse graphs that lighten the burden on convolutional operators, we encounter memory constraints when dealing with graph datasets containing a large number of nodes, on the order of $10^4$ nodes. To determine whether a connection should be established, we must calculate distances between all points starting from a pointcloud. However, this poses a challenge as the computational complexity scales quadratically with the number of nodes in the graph. Consequently, as the graph size increases, the computation quickly becomes intractable. To address the issue of potential memory overflows, we adopt Kernel Operations (KeOps) (Charlier et al., 2021), as recommended by previous studies on latent graph inference. KeOps enables us to perform computations on large arrays by efficiently reducing them based on a mathematical formula.

**Trainable Snowflake Activation Preliminary Experiments: Stability and Initialization.** Although in equation 1 the $\alpha$, $\beta$ and $\gamma$ parameters are introduced as trainable parameters for the sake of generality, we find that during backpropagation this exponential learnable terms can lead to instabilities, hence, we set them all to $\alpha = \beta = \gamma = 1$. In future research it could be explored how to stabilize these and whether this additional flexibility proves advantageous empirically. We run some initial experiments to assess the stability of the trainable snowflake activation. In particular we work with the homophilic benchmarks Cora and Citeseer and we incorporate the snowflake activation to dDGM with Euclidean space and which feeds its infered latent graph to a Graph Convolutional Network of 3 layers with ELU activation functions. That is, the snowflake activation takes as input the euclidean distance between latent graph nodes computed by the dDGM.

We observe that in this particular configuration the $p$ parameter that controls how much the quasi-metric deviates from being a metric can be a problematic parameter during training. Interestingly, this does not seem to be the case, when running synthetic experiments with a full neural snowflake (see Appendix I). This could be attributed to the fact that in the literature in this setup the dDGM is trained with a learning rate of $10^{-2}$, which is too aggressive to update the $p$ parameter. For Cora the dDGM leveraging Euclidean space and using the original dataset graph as inductive bias achieves an accuracy of $82.40 \pm 3.22$ (mean $\pm$ standard deviation), using the off-the-shelf snowflake activation we obtain $40.33 \pm 11.62$ which presents a clear drop in performance. The observed large standard deviation indicates that the model frequently becomes trapped in local minima during optimization. This can be mitigated by either setting $p = 0$ and learning a metric, or by using a different optimizer with a lower learning rate ($10^{-4}$, for example) to update the parameter $p$. This configurations lead to accuracies of $85.48 \pm 2.74$ and $86.11 \pm 3.72$ respectively for Cora, which clearly surpass the performance using Euclidean space. In the case of Citeseer using the original dDGM we get an accuracy of $73.40 \pm 1.64$, using a snowflake activation with $p = 0$ we obtain $73.85 \pm 2.34$, and using a learnable $p$ with a learning rate of $10^{-4}$ we achieve $74.40 \pm 2.08$. Note that for the cases in which we learn $p$ with a different optimizer, $p$ is initialized at $p = 10^{-8}$ to start from a metric and slowly learn a quasi-metric. These experiments show that the quasi-metric relaxation can provide

the snowflake activation with additional representation capabilities and flexibility but should be used with care. For all the rest of experiments in this paper we learn a quasi-metric when implementing the snowflake activation but use a slower optimizer for the $p$ parameter. In these experiments the coefficients $C_1$, $C_2$, and $C_3$ were initialized to 1. We use the absolute value function during training to ensure they stay non-negative. Note that these experiments were conducted using the Gumbel Top-k trick edge sampling algorithm with $k = 7$ and an embedding dimensionality of 4.

**Neural Snowflakes.** From a computational perspective, it is more reliable to assign fixed coefficients $a$ and $b$, instead of backpropagating through them. Specifically, we set $a = 1$ and $b = 1$ for all experiments. Matrices $A$, $B$, and $C$ weights are initialized by sampling from a uniform distribution ranging from 0 to 1 and normalized by the matrix dimensions. For example, in the case of $A$, the weights are sampled from a distribution between 0 and $1/(d_{A1}d_{A2})$, where $d_{A1}$ is the number of rows and $d_{A2}$ is the number of columns of the matrix. We experimentally observe that other initializations such as drawing the weights from a Gaussian or using Xavier initialization can lead to instabilities and exploding numbers in the forward pass. To guarantee non-negativity of all weights throughout training, we apply an absolute function activation to the weights. $p$ is initialized to $p = 1e - 8 \approx 0$, so that we start from a metric space and gradually learn a quasi-metric space. Furthermore, it should be noted that the learnable coefficient $p$ is exclusively applied in the last layer of the neural snowflake model. This design choice allows us to track the coefficient $C$, which represents the relaxation of the triangle inequality. In our synthetic experiments, we employ a readily available neural snowflake model, while for latent graph inference, we utilize a weighted skip connection. The neural snowflake model takes the Euclidean distance between latent representations as input. We observe that employing a skip connection and initiating training with an almost Euclidean metric proves beneficial, particularly during the early stages of training.

## I   EXPERIMENTAL RESULTS SUPPLEMENTARY MATERIAL

Within this appendix, we provide supplementary information regarding the experimental results discussed in the main text. This encompasses details about the train and test splits, the precise training configurations applied, as well as supplementary visual representations illustrating the evolution of the model training process, along with additional experiments and their corresponding results.

**Synthetic Experiments.** All models are trained using the Adam optimizer with a learning rate of $1 \times 10^{-4}$, for 40 epochs and with a batch size of 1,000. After approximately 20 epochs, we notice a tendency for learning to reach a plateau, particularly when dealing with neural snowflakes. As mentioned in the main text, for our experimental analysis, we concentrate on fully connected graphs. In this setup, the node coordinates are sampled randomly from a multivariate Gaussian distribution within a 100-dimensional hypercube in Euclidean space, represented as $\mathbb{R}^{100}$. The graph weights are determined based on the metrics outlined in Table 2. The training sets have 4,000,000 data points and the test sets 10,000. We observe very little discrepancy between the performance of the models for training and testing sets.

The MLP model, which works in isolation and aims at approximating the metrics using Euclidean space $\|\text{MLP}(\mathbf{x}) - \text{MLP}(\mathbf{y})\|$, has a total of 5422 parameters, and its composed of 10 linear layers with 20 hidden dimensions and ReLU activation functions. The neural snowflake learning the metric on $\mathbb{R}^2$: $f(\|\text{MLP}(\mathbf{x}) - \text{MLP}(\mathbf{y})\|)$, is composed of 2 layers with hidden dimension of 20. Note that in Section 3.1 we define $A^{(i)}$ as a $\tilde{d}_i \times d_{i-1}$ matrix, $B^{(i)}$ as a $d_i \times \tilde{d}_i$-matrix. However, in this experiments we set $\tilde{d}_i = d_i = 20$. The MLP used alongside the neural snowflake to project the node features in $\mathbb{R}^{100}$ to $\mathbb{R}^2$ consists of 5 layers with hidden dimension 20 with a total of 3,322 model parameters and also uses ReLU activations. Lastly, for the third case in which the neural snowflake learns directly in $\mathbb{R}^{100}$: $f(\|\mathbf{x} - \mathbf{y}\|)$ we reuse the same neural snowflake architecture as before with a total of 847 learnable parameters.

Additionally, we provide some plots of the training loss function evolution during learning for the synthetic graph embedding experiments in Figure 2. As we can observe from the plots the neural snowflakes learning in $\mathbb{R}^{100}$ converge faster, whereas neural snowflakes in $\mathbb{R}^2$ tend to get stuck in local minima at the beginning of training and eventually achieve comparable performance to their higher-dimensional counterparts. On the contrary, MLPs operating in $\mathbb{R}^2$ demonstrates significantly poorer performance, they achieve a higher loss with a higher variance during the training process. In

the main text we provided results for the synthetic experiments in terms of the test set performance; we additionally provide results for the training set in Table 6.

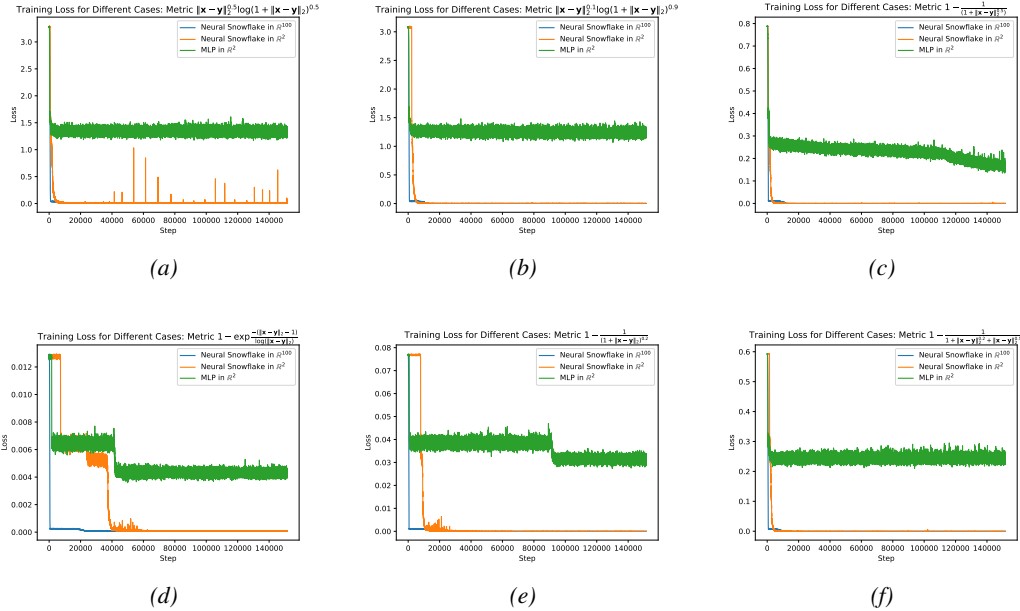

*Figure 2:* Training losses for synthetic graph embedding experiments. We compare using Euclidean space for encoding the weighted graphs to using snowflake quasi-metric spaces.

*Table 6:* Results for synthetic graph embedding experiments, mean square error for training set. The Neural Snowflake models are able to learn the metric better with substantially lesser number of model parameters.

| | **MLP** | **Neural Snowflake (+ MLP)** | **Neural Snowflake** |
|---|---|---|---|
| No. Parameters | 5422 | 4169 | 847 |
| Embedding space, $\mathbb{R}^n$ | 2 | 2 | 100 |
| Metric | | Mean Square Error | |
| $\|\mathbf{x} - \mathbf{y}\|^{0.5} \log(1 + \|\mathbf{x} - \mathbf{y}\|)^{0.5}$ | 1.3299 | 0.0037 | **0.0029** |
| $\|\mathbf{x} - \mathbf{y}\|^{0.1} \log(1 + \|\mathbf{x} - \mathbf{y}\|)^{0.9}$ | 1.2242 | 0.0032 | **0.0031** |
| $1 - \frac{1}{(1 + \|\mathbf{x} - \mathbf{y}\|^{0.5})}$ | 0.0777 | **0.00004** | 0.00004 |
| $1 - \exp \frac{-(\|\mathbf{x} - \mathbf{y}\| - 1)}{\log(\|\mathbf{x} - \mathbf{y}\|)}$ | 0.1649 | 0.00009 | **0.00008** |
| $1 - \frac{1}{(1 + \|\mathbf{x} - \mathbf{y}\|)^{0.2}}$ | 0.0314 | **0.00005** | 0.00005 |
| $1 - \frac{1}{1 + \|\mathbf{x} - \mathbf{y}\|^{0.2} + \|\mathbf{x} - \mathbf{y}\|^{0.5}}$ | 0.2420 | **0.00002** | 0.00002 |

**Snowflake activation.** Before working with neural snowflakes, we evaluate the performance of snowflake activations. This is a straightforward way of augmenting existing latent graph inference algorithms with additional representation power. In appendix H, we have already covered the details regarding the activation function's computational implementation. We intend to adhere to the specifications outlined in that section. We start by running some preliminary studies in which we compare the performance of a GCN equipped with a dDGM module using different latent embedding metric spaces. To ensure a fair comparison in terms of latent space geometry, we have set the dimensionality of the latent embedding space to 4. By keeping the dimensionality fixed, we ensure that we only modify the manifold used for embedding the representations, without altering the dimensionality of the embedding space itself. We start by evaluating the different algorithms on benchmark homophilic graph datasets such as Cora, CiteSeer, CS, and Physics. We start by comparing our model to previous metric spaces used in the literature such as Euclidean space (Kazi et al., 2022). In Table 7 we can observe that the snowflake activation helps achieve higher accuracies, leading to improvements between $1\% - 5\%$.

*Table 7:* Latent graph inference results across a variety of benchmark homophilic graph datasets. For all experiment the latent space dimensionality used to infer latent graphs is fixed to 4, and the Gumbel top-k trick is used with $k = 7$.

| Model | Metric Space | Input Graph | Cora | CiteSeer | CS | Physics |
|---|---|---|---|---|---|---|
| | | | Accuracy (%) $\pm$ Standard Deviation | | | |
| DGM | Euclidean | Yes | $82.40_{\pm3.22}$ | $73.40_{\pm1.64}$ | $85.45_{\pm2.23}$ | $95.91_{\pm0.51}$ |
| DGM | Snowflake | Yes | $86.11_{\pm3.72}$ | $74.40_{\pm2.08}$ | $89.54_{\pm2.48}$ | $96.08_{\pm0.46}$ |
| DGM | Euclidean | No | $62.03_{\pm6.20}$ | $65.15_{\pm4.84}$ | $84.37_{\pm1.20}$ | $95.11_{\pm0.33}$ |
| DGM | Snowflake | No | $67.33_{\pm4.10}$ | $64.63_{\pm9.24}$ | $87.63_{\pm5.25}$ | $95.32_{\pm0.45}$ |
| MLP | Euclidean | No | $58.92_{\pm3.28}$ | $59.48_{\pm2.14}$ | $87.80_{\pm1.54}$ | $94.91_{\pm0.30}$ |

Next, we increase the dimensionality of the latent space from 4 to 8 and evaluate whether this trend still persists. In Table 8, we can observe that as we increase the dimensionality of the latent space used for inferring the latent graph the performance increases when using both Euclidean or snowflake quasi-metric spaces. We can also see that the difference between the two becomes less significant, except for the CS dataset in which the snowflake activation still performs substantially better.

*Table 8:* Latent graph inference results with latent space dimensionality fixed to 8, and the Gumbel top-k trick is used with $k = 7$.

| Model | Metric Space | Input Graph | Cora | CiteSeer | CS | Physics |
|---|---|---|---|---|---|---|
| | | | Accuracy (%) $\pm$ Standard Deviation | | | |
| DGM | Euclidean | Yes | $85.77_{\pm3.64}$ | $73.67_{\pm2.30}$ | $90.50_{\pm1.89}$ | $96.08_{\pm0.41}$ |
| DGM | Snowflake | Yes | $85.41_{\pm3.70}$ | $74.19_{\pm2.08}$ | $92.98_{\pm0.66}$ | $96.15_{\pm0.54}$ |
| DGM | Euclidean | No | $68.37_{\pm5.39}$ | $68.10_{\pm2.80}$ | $88.17_{\pm2.64}$ | $95.27_{\pm0.41}$ |
| DGM | Snowflake | No | $69.51_{\pm4.42}$ | $66.86_{\pm2.82}$ | $88.67_{\pm3.21}$ | $95.36_{\pm0.23}$ |
| MLP | Euclidean | No | $58.92_{\pm3.28}$ | $59.48_{\pm2.14}$ | $87.80_{\pm1.54}$ | $94.91_{\pm0.30}$ |

In the specific scenario presented in Table 9, we observe a contrasting effect. As the latent space dimension is reduced to only 2, the performance of both models deteriorates. Nonetheless, it is notable that the use of the snowflake metric space demonstrates greater resilience compared to its Euclidean counterpart. In fact, we can observe performance discrepancies of up to 20% between the two. This is in line with previous synthetic experiments, and demonstrates that snowflake quasi-metric spaces are more efficient at compressing the same information in low-dimensional spaces.

*Table 9:* Latent graph inference results with latent space dimensionality fixed to 2, and the Gumbel top-k trick is used with $k = 7$.

| Model | Metric Space | Input Graph | Cora | CiteSeer | CS | Physics |
|---|---|---|---|---|---|---|
| | | | Accuracy (%) $\pm$ Standard Deviation | | | |
| DGM | Euclidean | Yes | $59.25_{\pm14.60}$ | $70.00_{\pm2.15}$ | $62.15_{\pm2.92}$ | $92.01_{\pm2.74}$ |
| DGM | Snowflake | Yes | $79.44_{\pm6.50}$ | $69.51_{\pm3.95}$ | $79.75_{\pm2.57}$ | $94.29_{\pm2.91}$ |
| DGM | Euclidean | No | $42.40_{\pm8.20}$ | $60.93_{\pm4.07}$ | $69.93_{\pm2.17}$ | $86.05_{\pm2.77}$ |
| DGM | Snowflake | No | $64.18_{\pm3.46}$ | $64.61_{\pm6.14}$ | $81.70_{\pm5.35}$ | $93.45_{\pm2.93}$ |
| MLP | Euclidean | No | $58.92_{\pm3.28}$ | $59.48_{\pm2.14}$ | $87.80_{\pm1.54}$ | $94.91_{\pm0.30}$ |