# OpenReview forum: "Neural Snowflakes: Universal Latent Graph Inference via Trainable Latent Geometries"
_ICLR.cc/2024/Conference — ICLR 2024 poster_

### Official Review · Reviewer_cbgk · 2023-10-27

**Soundness:** 3 good
**Presentation:** 3 good
**Contribution:** 3 good
**Rating:** 6
**Confidence:** 2

**Summary:**

The authors present Neural Snowflakes, a latent graph inference method. Neural Snowflakes are iterative representations that use a trainable activation function to refine input distance values $\|x - y\|$. The authors present theoretic justification for their model as well as compelling empirical results on latent graph inference tasks.

**Strengths:**

* The authors present compelling empirical results for their method
* Theoretical analysis underpins these empirical results
* The fact that Neural Snowflakes can break the curse of dimensionality and don't need exponentially many parameters is a very nice property

**Weaknesses:**

* The paper does not state the loss function the Neural Snowflake models are trained on
* There is no mentioning of the computational complexity/ scalability of the approach: does it scale to hundreds of thousands/ millions of data points? Why (not)?

**Questions:**

Admittedly, I am not an expert on latent graph inference, so I'm doing my best to judge the quality of the work based on the description by the authors and the results produced.

---

> ### Author Response · Authors · 2023-11-16
> **Clarifications and Scalability**
>
> We thank the reviewer for the questions. We address them below.
>
> **“The paper does not state the loss function the Neural Snowflake models are trained on”**
>
> We have addressed this concern in Appendix F: we use a combination of a cross-entropy loss and a graph loss, in line with previous research (Differentiable Graph Module Kazi et. al).
>
> **“There is no mentioning of the computational complexity/ scalability of the approach: does it scale to hundreds of thousands/ millions of data points? Why (not)?”**
>
> Please refer to Appendix H for a detailed discussion on computational complexity and scalability. In essence, our model, the neural snowflake, is computationally inexpensive. The primary computational complexity is derived from the dDGM module. However, our theoretical analysis and quasi-metric learning model could be integrated into more scalable latent graph inference pipelines, such as the NodeFormer. A paragraph has been added to the conclusion of the paper to address this aspect. It's important to note that our emphasis is on the geometric structure of the latent embedding spaces used for inferring edges, rather than on enhancing the computational complexity or scalability of existing models.
>
> Please, let us know if we could address any further concerns.

---

> > ### Comment · Reviewer_cbgk · 2023-11-21
> >
> > Dear authors,
> >
> > Thanks for addressing my comments. I will keep my original score for now.

---

### Official Review · Reviewer_pN6S · 2023-10-30

**Soundness:** 4 excellent
**Presentation:** 4 excellent
**Contribution:** 4 excellent
**Rating:** 8
**Confidence:** 3

**Summary:**

The paper presents and analyzes a trainable metric termed "neural snowflake", which is a combination of a bounded gaussian kernel distance , a fractal component $\vert a-b \vert^\alpha$ with $\alpha \in (0,1]$ and an *irregular* fractal component in log space $\log\left(1+\Vert a-b\Vert^\beta\right)^{1+\vert p \vert}$ with $\beta \in (0,1],p\in \mathbb{R}_0^+$. This distance function is used to learn *latent* graphs from data, wherein the graph structure of a dataset or sample is inferred from the features, instead of being constructed by a human annotater.

The paper proves that

1. parametrized with an MLP as embedder and the neural snowflake and distance function, we can learn a $d$ dimensional embedding for any finite weighted graph
2. There are finite weighted graphs which are *not* similarly representable if using a euclidean distance
3. the depth and hidden width of the MLP depends favourably on the number of nodes and embedding dimensions (order $(n\log n)^{1.5}$ and linearly, respectively)
4. For specific graphs (e.g. trees) even more favourable guarantees are possible

The method is evaluated on Cora, CiteSeer (standard GNN benchmarks) and Tadpole and Aerothermodynamics (used in the paper which introduced the latent graph approach.)

**Strengths:**

The paper is very accessible for such a technical topic and presents its story clearly and convincingly.


Novelty/Originality: Moving from euclidean and Riemanian metrics to quasi-metric spaces is an idea which I haven't seen a lot in the literature ...and I checked because I was actually thinking about going into this direction myself but hadn't properly started working on this due to previous projects. Happy to be "scooped" in this manner!

Significance: Having provable representation power in this sense for arbitrary graphs will be an important ingredient as we move towards building less supervised relational ML models

Quality: derivations, expositions and experiments are well presented and clean. Proofs look sane to me as well, although I did not step through them in detail

**Weaknesses:**

1.  There is relatively high variances in most of the tables, could you please perform suitable significance tests (e.g. with a bonferonni correction) and mark those that are significant to aid the viewer?
2. I might have missed it, but what is the STD across? Evaluations or trainings? E.g. in table 7
3. Nitpick, but explicitly discussing the memory constraints and other concerns in a limitations section in the appendix helps the readers judge potential pitfalls

**Questions:**

Aside from Q2 in weakness, no questions

---

> ### Author Response · Authors · 2023-11-16
> **Additional Clarification on Numerics**
>
> We express our gratitude to the reviewer for their insightful comments, which have contributed to enhancing the quality of our manuscript. A revised version has been submitted to OpenReview, where the reviewer can observe the incorporation of many of their suggestions. Our responses to their comments are provided below.
>
> **“There is relatively high variances in most of the tables, could you please perform suitable significance tests (e.g. with a bonferonni correction) and mark those that are significant to aid the viewer?”**
>
> Thank you for bringing this to our attention. Regrettably, given the time constraints during the rebuttal period, conducting significance tests for all experiments poses a considerable challenge. Nevertheless, it's worth noting that previous papers like "Differentiable Graph Module" and "Latent Graph Inference using Product Manifolds" have reported similar variances. We hope this provides some indication of the reliability of our experimental results.
>
> **“I might have missed it, but what is the STD across? Evaluations or trainings? E.g. in table 7”**
> We report results for the test set. We use the same setup as the Differentiable Graph Module paper by Kazi et. al for consistency, 10 fold cross-validation.
>
>
> **“Nitpick, but explicitly discussing the memory constraints and other concerns in a limitations section in the appendix helps the readers judge potential pitfalls”**
>
> We agree discussions on memory constraints and other limitations are important. Please see Appendix H in which these are discussed. Note that computational limitations are inherited from the dDGM module, whereas in principle, our quasi-metric learning neural snowflake is computationally efficient. We have also added some text to the conclusion discussing scalability to larger graphs and future research directions.
>
> Please let us know if we can address any further concerns.

---

> > ### Comment · Reviewer_pN6S · 2023-11-22
> >
> > Thank you for your comments. I recommend using the time until the camera ready to perform significance tests, maybe following the reasoning laid out in https://www.jmlr.org/papers/volume7/demsar06a/demsar06a.pdf.

---

### Official Review · Reviewer_gXAo · 2023-10-31

**Soundness:** 3 good
**Presentation:** 2 fair
**Contribution:** 2 fair
**Rating:** 5
**Confidence:** 3

**Summary:**

This work studies learning latent structures from pure point clouds (a set of points without graph structures) or graph-structured data (a set of points with observed structures). The authors present rogirous analysis into the property of latent structure learning models and how to conduct latent graph inference that satifies the metric properties. Experiments on several benchmark datasets verify the proposed approach.

**Strengths:**

1. The paper studies an important and interesting problem

2. The theoretical analysis is rigorous and with technical depth

3. The overall writing is good though some parts can be improved to increase the readability

**Weaknesses:**

1. One concern lies in the clarity of the proposed model. The paper title describes the model as a "universal" method, but the main paper does not justify this point very well. For example, how universal the model is and to what extent? What is the advantage of the proposed model compared with prior art? Also, the algorithm for the model is missing, which makes the reader hard to understand what is precisely done in the model implementation.

2. The implication of the theoretical results needs to be made more clear. How can the theory apply to practical problems and what is the insight behind the analysis?

3. The experiments are limited with four small datasets. For the Cora and Citeseer, it seems that the common benchmark settings are not used for evaluation. Can the authors provide more reasons on this and what precisely the data splits are used for the present experiments? The baselines for comparison are not sufficient, and some of the typical graph structure learning models are missing, e.g., LDS [1] and the state-of-the-art NodeFormer [2] (that also considers the Gumbel trick).

[1] Luca Franceschi et al., Learning discrete structures for graph neural networks. In ICML, 2019.

[2] Qitian Wu et al., NodeFormer: A Scalable Graph Structure Learning Transformer for Node Classification. In NeurIPS, 2022.

**Questions:**

See some of the questions in the weakness section. Besides, there are some further questions:

1. Can the model handle more difficult datasets, e.g., heterophilic graphs and graphs with incomplete edges?

2. How does the model compare with recent graph structure learning models LDS [1] and NodeFormer [2]?

3. What is the impact of the informativeness of node features on the model performance when not using the input graph?

4. Can the theoretical results be applied to graph Transformers whose attention networks can be seen as latent graph inference?

---

> ### Author Response · Authors · 2023-11-16
> **Universal Graph Embedding Its Implications on Provable Latent Graph Inference**
>
> **"One concern lies in the clarity of the proposed model. The paper title describes the model as a "universal" method, but the main paper does not justify this point very well. For example, how universal the model is and to what extent?”**
>
> Thank you for bringing this to our attention. We have made revisions to the manuscript to enhance the clarity of our contribution. Specifically, we have formalized universal graph embedding. Additional content has been incorporated into Section 4.1.1, including Definition 1 (Universal Graph Embedding) and Theorem 1 (Generic Graph Reconstruction via Universal Graph Inference Models). Please consult Appendix C.2 for the proof of Theorem 1.
>
> The material presented in Section 4.1.1 provides a comprehensive understanding of the desired properties for a latent graph inference model. Subsequently, Theorem 2 demonstrates that, for any finite weighted graph $G$, the neural snowflake model can establish a geometry in $\mathbb{R}^d$, allowing a simple MLP encoder to isometrically embed $G$. Thus, Theorem 2 serves as a universal embedding theorem aligned with our specified requirements. Note that this is distinct from more conventional universal approximation theorems.
>
> **“What is the advantage of the proposed model compared with prior art?”**
>
> In this work we are comparing models in terms of their latent space geometric structure. The current state of the art in embedding-based latent graph inference involves embedding into product Riemannian manifolds and inferring edges using differentiable k-nearest neighbor algorithms (as discussed in "Latent Graph Inference using Product Manifolds, ICLR 2023"). In addition to the experimental comparisons, our primary theoretical analysis is provided by Proposition 2. This proposition demonstrates that even a simple graph cannot be isometrically embedded into any Riemannian manifold, including the product manifolds utilized in the current state-of-the-art approaches.
>
> Upon comparison with our universal embedding theorem, specifically Theorem 2, we observe a theoretical advantage. This is because our proposed model can isometrically embed any graph, presenting a contrast to the limitations highlighted by Proposition 2 in the context of existing Riemannian manifolds, including those employed in state-of-the-art methods.
>
> **“Also, the algorithm for the model is missing, which makes the reader hard to understand what is precisely done in the model implementation.”**
>
> Thank you for your suggestion. The mathematical description of the neural snowflake architecture can be found in Section 3.1. However, in response to the reviewers' suggestions, we have included an algorithmic description of the model in Appendix G. We hope this provides clarity regarding our implementation. Additionally, further computational implementation details, including hardware specifications, the utilization of symbolic matrices, and hyperparameters, are discussed in Appendix H.
>
> **“The implication of the theoretical results needs to be made more clear. How can the theory apply to practical problems and what is the insight behind the analysis?”**
>
> We agree and have added a new theorem, Theorem 1, which clarifies precisely this point.  Briefly, it states that any universal latent graph inference model, as now defined clearly in Definition 1, is such that our inference algorithm can reconstruct the graph from an embedding implementable by the neural snowflake + MLP graph inference model.  Therefore, together Theorem 1 and our universal embedding theorem, namely Theorem 2, imply that graph reconstruction is possible with the neural snowflake precisely since it can isometrically embed any graph.

---

> ### Author Response · Authors · 2023-11-16
> **Details on Experiments**
>
> **“The experiments are limited with four small datasets. For the Cora and Citeseer, it seems that the common benchmark settings are not used for evaluation. Can the authors provide more reasons on this and what precisely the data splits are used for the present experiments? The baselines for comparison are not sufficient, and some of the typical graph structure learning models are missing, e.g., LDS [1] and the state-of-the-art NodeFormer [2] (that also considers the Gumbel trick).” “How does the model compare with recent graph structure learning models LDS [1] and NodeFormer [2]?” “Can the theoretical results be applied to graph Transformers whose attention networks can be seen as latent graph inference?”**
>
> We have addressed these concerns by adding a paragraph to the conclusion. Our primary research goal was to compare the representation capabilities of different embedding spaces. To ensure a fair comparison, we maintained a fixed architecture for the latent graph inference module and solely modified the latent geometry. Models like the NodeFormer primarily emphasize achieving scale, focusing less on analyzing the geometric structure of the latent space used for inferring the latent graph.
> Specifically, the NodeFormer architecture, in its effort to expedite computation, refrains from directly comparing node feature representations. Instead, it utilizes an averaged embedded quantity to compute similarities. Consequently, making a straightforward comparison in terms of latent space structure and latent graph inferability capabilities becomes challenging and could potentially be misleading. We posit that integrating the geometric concepts discussed in our work with scalable architectures like NodeFormer, rather than DGM, presents an intriguing avenue for future research.
> In terms of data splits we use the same setup as the Differentiable Graph Module paper by Kazi et. al for consistency, 10 fold cross-validation.
>
>
> **“Can the model handle more difficult datasets, e.g., heterophilic graphs and graphs with incomplete edges?”**
>
> The model can function even in the absence of edges, allowing it to operate effectively with incomplete edges. Furthermore, the DGM module has demonstrated its effectiveness in handling heterophilic graphs in prior research (as detailed in "Latent Graph Inference using Product Manifolds, ICLR 2023").
>
> **“What is the impact of the informativeness of node features on the model performance when not using the input graph?”**
>
> In general, performance tends to degrade without the input graph. The way the latent graph inference module leverages node features depends on the GNN diffusion layers employed downstream, as both are optimized in parallel. For instance, when utilizing GCN layers, the model aims to construct a homophilic graph that GCN layers can efficiently diffuse information on by connecting nodes with similar feature vectors.
>
>
> Please, let us know if we could address any further concerns.

---

### Official Review · Reviewer_Sjjv · 2023-11-16

**Soundness:** 4 excellent
**Presentation:** 4 excellent
**Contribution:** 4 excellent
**Rating:** 8
**Confidence:** 2

**Summary:**

The authors leverage the representation power of a snowflake metrics using learnable neural snowflake activation in a neural network-based model. In doing so, they prove that neural snowflake permits universal representation for learning latent graphs, which is not the case for simpler previous state-of-the-art geometric embeddings. This is demonstrated theoretically via Theorems 1-2, with particular examples provided to motivate a learnable, non-Riemannian metric. This is also demonstrated empirically with a comprehensive comparison to alternative latent graph inference methods which use more typical metrics, clearly outperforming these baselines in some cases, and performing competitively in others.

**Strengths:**

1. The benefit of the extension to quasi-metric spaces and a learnable neural snowflake activation is clearly explained, as the authors provide both theoretical findings and concrete examples to demonstrate the utility.
2. The experiments are thorough and well-controlled. The striking improvement in applying neural snowflake in the synthetic setting is convincing, and the marginal benefit in the "input graph" experiment of Table 3 demonstrates that this advantage is present in real data as well.
3. The universal representation power in conjunction with the fact that the model provably needs not to use a large number of parameters is a very strong theoretical result.

Overall, this is a very well-written and convincing work. While I am not entirely knowledgeable on the topic of latent graph inference, I think that the theoretical and empirical arguments of the authors provide a very clear explanation of the novelty of their approach and how it fits in the context of the current state of the field.

**Weaknesses:**

While the majority of the experimental results are convincing, the results of Table 4 are slightly underwhelming. It is certainly a good finding that neural snowflake is competitive in all cases, which cannot be said about any of the other approaches, but the fact that it is usually outperformed by some other method limits the applicability of this approach to real data.

Maybe there is some experiment to evaluate the method in a way that provides more initial information than the experiment in Table 4, but not entirely the original input graph as used in Table 3? However, I don't believe this is entirely necessary to convey the utility of the approach.

**Questions:**

While you note that this is mainly tangential to the primary focus of this paper, you mention that the suboptimality of the Gumbel Top-k edge sampling algorithm may be a reason why the improved performance of your method in the real data settings is not as pronounced. However, would this not affect all methods equally? That is, why do you suggest that an improvement in this regard might better differentiate the neural snowflake model from the baselines?

---

> ### Author Response · Authors · 2023-11-21
> **Gumble Top-k Homogenizes Performance Across Latent Graph Inference Models**
>
> Dear reviewer, we would like to thank you for your great review and your very interesting questions.
>
>
> **While the majority of the experimental results are convincing, the results of Table 4 are slightly underwhelming. It is certainly a good finding that neural snowflake is competitive in all cases, which cannot be said about any of the other approaches, but the fact that it is usually outperformed by some other method limits the applicability of this approach to real data.**
>
> That is a fair point.  However, unlike the benchmarked methods, each of whose performance varies on each graph inference task the neural snowflake offers consistent top 2 performance.  In particular, this means one can train a single neural snowflake model “out of the box” and achieve competitive performance without having to search through all the representation-space options when performing latent graph inference, which is computationally costly to achieve only slightly better results in some cases.
>
>
> **Maybe there is some experiment to evaluate the method in a way that provides more initial information than t
> he experiment in Table 4, but not entirely the original input graph as used in Table 3? However, I don't believe this is entirely necessary to convey the utility of the approach.**
>
> This would serve as a compelling evaluation metric. While we currently lack awareness of a specific metric or have one in mind, the prospect of devising one for future research is intriguing. Randomly masking certain graph edges could potentially offer a suitable testing method. Thank you for the suggestion, we will consider this in future research.
>
>
>
> **While you note that this is mainly tangential to the primary focus of this paper, you mention that the suboptimality of the Gumbel Top-k edge sampling algorithm may be a reason why the improved performance of your method in the real data settings is not as pronounced. However, would this not affect all methods equally? That is, why do you suggest that an improvement in this regard might better differentiate the neural snowflake model from the baselines?**
>
>
> Certainly, the Gumbel top-k trick affects all embedding spaces, not only the neural snowflake. In the synthetic experiments, we focused on metric learning, in which we showed that neural snowflakes excel. Unlike the latent graph inference experiments, these results are unaffected by edge sampling. In the latent graph inference experiments, the metric learned by the embedding spaces can get ‘lost in translation’ when ‘translating’ the learned continuous representation into a discrete graph. The Gumbel top-k trick samples k edges for every node irrespective of the embedding space: this means that the sampled graphs end up having very similar characteristics. The key observation is that the embedding spaces, in the context of latent graph inference, exhibit similar performance because the sampled graphs are consistently similar, regardless of the representation capabilities of the embedding spaces they stem from. Hence, given that the neural snowflake performance is much better in its continuous form (as shown in the synthetic experiments), in relative terms, its performance degrades more when combined with the Gumbel top-k trick for latent graph inference.

---

### Author Response · Authors · 2023-11-16
**Thanks for Improving the Quality of the Manuscript**

We thank the reviewer for their thoughtful comments, which will improve the quality of our manuscript. We have uploaded a revision to OpenReview, where the reviewer will find that we have incorporated many of their suggestions. We respond to their comments below.

We would also like to draw the reviewer's attention to the newly added Theorem 1, in response to one of the referee's questions.  The new result shows that any universal graph inference model, such as the proposed neural snowflake + MLP model, indeed has the power to reconstruct a graph from its node embeddings by connecting nearest neighbours.  We feel that this result clarifies the power of our universal graph embedding theorem, namely Theorem 2.

We want to thank all the reviewers again for their very helpful feedback and insights.  We're very happy with how this has influenced and strengthened our research paper.

---

### Meta-Review · Area_Chair_eiuA · 2023-12-15

**Metareview:**

The paper introduces a novel approach called "neural snowflake" for learning latent graphs using a learnable neural snowflake activation in a neural network-based model, which has the potential for less supervised relational machine learning models, and adopts an innovative snowflake activation function, demonstrates strong theoretical results in sowing the universal representation power. Although the paper might fall short in terms of empirical breadth and addressing practical constraints and scalability, the paper makes a substantial contribution to the field, especially in the theoretical understanding and practical application of latent graph learning.

**Justification For Why Not Higher Score:**

The paper's empirical evaluation is somewhat limited.

**Justification For Why Not Lower Score:**

The novelty of the proposed approach is clear.

---

### Decision · Program_Chairs · 2024-01-16

Accept (poster)